# A Compressive-Expressive Communication Framework for Compositional Representations

**Rafael Elberg** [*]
Pontificia Universidad Católica, CENIA, i-Health
Chile
rafael.elberg@uc.cl

**Felipe del Rio**
Pontificia Universidad Católica, CENIA
Chile
fidelrio@uc.cl

**Mircea Petrache**
Pontificia Universidad Católica, CENIA
Chile
mpetrache@uc.cl

**Denis Parra**
Pontificia Universidad Católica, CENIA, i-Health
Chile
dparras@uc.cl

## Abstract

Compositionality in knowledge and language—the ability to represent complex concepts as a combination of simpler ones—is a hallmark of human cognition and communication. Despite recent advances, deep neural networks still struggle to acquire this property reliably. Neural models for emergent communication look to endow artificial agents with compositional language by simulating the pressures that form human language. In this work, we introduce CELEBI[2] (Compressive-Expressive Language Emergence through a discrete Bottleneck and Iterated learning), a novel self-supervised framework for inducing compositional representations through a reconstruction-based communication game between a sender and a receiver. Building on theories of language emergence and the iterated learning framework, we integrate three mechanisms that jointly promote compressibility, expressivity, and efficiency in the emergent language. First, *Progressive Decoding* incentivizes intermediate reasoning by requiring the receiver to produce partial reconstructions after each symbol. Second, *Final-State Imitation* trains successive generations of agents to imitate reconstructions rather than messages, enforcing a tighter communication bottleneck. Third, *Pairwise Distance Maximization* regularizes message diversity by encouraging high distances between messages, with formal links to entropy maximization. Our method significantly improves both the efficiency and compositionality of the learned messages on the Shapes3D and MPI3D datasets, surpassing prior discrete communication frameworks in both reconstruction accuracy and topographic similarity. This work provides new theoretical and empirical evidence for the emergence of structured, generalizable communication protocols from simplicity-based inductive biases.

## 1 Introduction

In natural languages, compositionality enables humans to communicate an infinite number of ideas using a finite set of elements [12]. This principle enables speakers to flexibly combine known words and structures to convey novel meanings, supporting flexible and generalizable communication across diverse and previously unseen contexts.

---

[*]Corresponding author

[2]Our oficial implementation can be found at https://github.com/SugarFreeManatee/CELEBI

39th Conference on Neural Information Processing Systems (NeurIPS 2025).

Works on the emergence of compositionality in language [34, 35, 37] have argued that opposing pressures are necessary for the natural selection of compositional languages between generations of speakers [20, 3]. On the one hand, successful communication requires **high expressivity** in order to usefully describe the world, allowing speakers to produce distinct messages for a wide range of meanings. On the other hand, models of emergent communication such as the iterated learning framework [34] state that natural speakers tend to minimize the complexity of languages through cultural transmission, implying that **simpler and more compressive languages** are more easily passed on to new speakers.

These opposite pressures of expressivity and compressibility thus generate a trade-off, highlighted in several works [35, 20, 3] which is argued to be optimized by compositional languages, whereby languages evolve to maximize communicative efficiency – remaining expressive enough to convey diverse meanings while being simple and structured enough to be easily learned and transmitted by successive generations of speakers. When either pressure is dominant, undesirable languages tend to emerge: (a) excessive language compression leads to *degenerate* languages where multiple meanings are mapped to the same messages making them ambiguous and thus hard to use, and (b) prioritizing expressivity alone produces in *holistic* languages i.e. languages where messages are mapped arbitrarily to meanings without respecting their structures, hindering their transmission across generations [55].

Enabling machines to generalize compositionally is thought to be crucial for them to quickly adapt to novel situations beyond their training experience [42]. Inducing compositional behavior in neural networks remains a major challenge [39, 30, 59, 33]. A growing body of work suggests that a model's ability to generalize compositionally is highly sensitive to its training conditions [40, 14, 60, 41, 15], including factors such as data distribution, learning objectives, and task design. One promising direction is the study of emergent communication, where discrete languages evolve for coordination between independent neural agents.

In this work, we build on the Lewis reconstruction game framework [58] to study emergent communication in a reconstruction task. A sender encodes an image into a discrete message and a receiver reconstructs the original. Drawing on theories of language evolution [34, 35], we develop a novel framework for inducing compositional communication grounded in simplicity bias.

We introduce CELEBI (Compressive-Expressive Language Emergence through a discrete Bottleneck and Iterated learning). Within this framework, we introduce three mechanisms in the learned communication protocol:

**Progressive Decoding**, in which the receiver makes reconstruction predictions after each incoming symbol, rather than waiting until the full message is received. This biases the system towards using intermediate reasoning steps, thereby imposing a pressure towards lower-complexity and less holistic encodings. While this mechanism does not directly improve reconstruction accuracy, it yields more efficient and structured communication.

**Final-State Imitation** modifies the standard imitation phase in the iterated learning (IL) framework of cultural evolution [35–37]. Instead of imitating the entire message, the student is trained to reproduce only the final output of the receiver, effectively reducing the information transmitted across generations. This tighter generational bottleneck increases pressure for compressibility in the emergent communication protocol [64], thereby promoting the emergence of compositional structure as a necessary condition for successful transmission. Empirically, this leads to increased compositionality with minimal degradation in reconstruction accuracy.

**Pairwise Distance Maximization** as a regularization term, i.e., as a further pressure towards increased diversity in the emergent language, that encourages exploration during the imitation phase. This term pushes the student sender to maximize an approximate Hamming distance between messages in a given batch, finding the most "diverse" protocol. We prove that this regularization gives a lower bound for entropy maximization [69], and an upper bound on the contrastive learning loss NT-Xtent [11].

The opposing pressures induced by these three mechanisms impose a tight regularization on the complexity of the emergent language. These ingredients make explicit the pressures proposed in the cognitive science and linguistic literature to shape language evolution, thus aligning with the IL paradigm and leading to measurably higher compositionality.

## 2 Background

### 2.1 Problem setup: recovering compositional representations

In a formalism similar to [56], we model a dataset $\mathcal{D} = \{x = \mathsf{GenX}(\mathbf{G}) : \mathbf{G} \in \mathcal{G}\}$, i.e. data $x$ are created via a deterministic function $\mathsf{GenX}$ from a set $\mathcal{G}$ of generating factors $\mathbf{G}$. We have access to samples from $\mathcal{D}$ but not to $\mathsf{GenX}$ or to the structure of $\mathbf{G}$, and the goal is to reproduce the set of $x$, i.e., generate images $\widehat{x}$ that approximate the distribution of $x$ in a natural metric such as $\mathbb{E}_{x \in \mathcal{D}}[\mathrm{MSE}(x, \widehat{x})]$, where MSE is the mean squared error on pixels.

The factors $\mathbf{G}$ are assumed to have compositional structure of the form $\mathbf{G} = [G_1, \ldots, G_n]$, in which $G_i$ represent independent characteristics having finitely many possible values (as in the dataset of Sec. 4.1 for example). In this case, an efficient way to reproduce elements of $\mathcal{D}$ is to impose that $\widehat{x}$ is similarly generated as $\widehat{x} = \mathsf{GenX}'(\mathbf{G}')$, with generating factors $\mathbf{G}'$ giving a learned encoding of $\mathbf{G}$.

As $\mathcal{G}$ has a large number of classes (one per combination of factor values) and we have a single example per class, successful reconstruction relies on the compositional structure of $\mathbf{G}$. Thus the goal is to **build a framework for finding the optimized compositional encoding $\mathbf{G}'$ from observation of a small training set** $\mathcal{D}_{train} \subset \mathcal{D}$. The compositional nature of $\mathbf{G}$ is what makes reconstruction possible from a small random $\mathcal{D}_{train}$, i.e. from data corresponding to a small subset of classes. This would be of course impossible for non-compositional data (see §D for a formal treatment).

### 2.2 Approach: Lewis reconstruction game

We frame the above reconstruction problem in the form of a cooperative game between two agents: a sender $S$ and a receiver $R$. The goal of $S$ will be to build the reconstruction factors $\mathbf{G}'$ and the receiver $R$ will map the factors to an image $\widehat{x}$. The two agents aim to create a protocol that reconstructs $x$ to good accuracy. This setup fits in the general class of Lewis Reconstruction Games (LRG) [58], a subclass of so-called Lewis Signaling Games (LSGs) [43].

Within the framework of signaling games, the factors $\mathbf{G}'$ are interpreted as a language which $S$ uses to communicate $x$ to $R$ as accurately as possible. The allowed message space is finite, i.e. $\mathbf{G}' \in \mathcal{M} := V^C$, and we take $C$ fixed and large enough that $|V|^C \gg |\mathcal{G}|$, so that the size of the message space is not a limiting factor on the emergent communication. The following diagram summarizes our notations:

$$\mathcal{G} \xrightarrow{\mathsf{GenX}} \mathcal{D} \subseteq \mathcal{X} \xrightarrow{S} \mathcal{M} = V^C \xrightarrow{R} \mathcal{X}$$

Our main focus is on defining a IL process which favors the emergence of a compositional language between $S$ and $R$ that accurately recovers $x \in \mathcal{D}$.

### 2.3 Base model architecture

We adopt a standard emergent communication setup as described by Xu et al. [71], see implementation details in § B.1. Sender and Receiver are implemented using the EGG framework [32]. Input images are encoded into latent representations using a pretrained VAE visual backbone. The sender encodes this latent input into a discrete message using an LSTM and then transmits it through a fixed-length communication channel. Message generation is made differentiable via a Gumbel-Softmax bottleneck [26]. The receiver takes the full message as inputs in an LSTM, and produces a latent vector, which is subsequently decoded into a reconstructed image by the VAE decoder. Reconstruction loss is computed between the original and generated images using Mean Squared Error (MSE). Both VAE encoder and decoder remain frozen during training.

## 3 Methods

This section describes our proposed improvements to the EL framework [71] (cf. Figure 1a). Intuitively, our additions attempt to increase the language drift between iterations of speakers, in order to thoroughly explore the landscape of possible languages, while maintaining efficient and useful communication. For the latter, we introduce **Progressive Decoding (PD)**, to serve as an anchor for the emergent language during interaction, and for the former we introduce **Final-State Imitation (FiSI)** and **Parwise Distance Maximization (PDM)**, which aim to incentivize exploration during the

imitation phase. We expect the combination of these factors to show the strongest impact towards compositionality in the learned language, more so than the factors taken separately.

## 3.1 Progressive Decoding (PD)

Our first proposed enhancement to the baseline communication protocol aims to improve the efficiency of the emergent language determined by the Sender and Receiver, by condensing information in fewer message tokens. This type of efficiency has been argued to be necessary for the emergence of natural language properties [29, 49].

Standard reconstruction games typically optimize a reconstruction loss computed only at the end of the communication phase, after the receiver has observed the full message [58], and as shown by Rita et al. [57], without an incentive to efficiently use the information received in the message, the receiver only updates its internal state after the full message has been received. To improve communication efficiency, we propose the following *Progressive Decoding (PD)* objective: the receiver generates a prediction after each received symbol, and our interaction phase loss explicitly weights the reconstruction error of each sub-message, as follows.

$$
\mathcal{L}_{\omega,\phi}^{\text{int}} := \mathbb{E}_x \left[ \frac{1}{C} \sum_{i=1}^{C} \lambda^i \, \text{MSE} \left( x, R_\omega(S_\phi(x)_{[i]}) \right) \right], \tag{1}
$$

where $S_\phi(x)_{[i]}$ is the prefix of length $i$ of the sender's message, and $\lambda \geq 1$ is an expressivity hyperparameter that increases the weight of later reconstructions. This loss has the following effects:

**Efficiency pressure:** Loss (1) rewards accurate reconstructions made as early as possible, favoring messages with shorter useful length.

**Interpretable sub-messages:** The model is encouraged to structure messages such that an interpretation by $R_\omega$ is available for sub-messages, making each symbol accountable for partial reconstruction. This echoes findings in prior theoretical and empirical work on chain-of-thought and compositionality [66, 28, 10, 52, 68, 54], which show that forcing intermediate semantic consistency or intermediate reasoning steps encourages models to show increased compositional behavior.

**Tuning the efficiency pressure:** Small $\lambda$ yields fast but potentially coarse reconstructions, while larger values favor detailed outputs at the cost of longer, possibly redundant messages, and in the limit $\lambda \to \infty$ (1) becomes equivalent to using just the last term in brackets, i.e., the full-length message reconstruction. In our experiments (see Section 5.2), we explore this trade-off empirically and show that $\lambda \equiv 1.5$ achieves the best performance across generalization, communication efficiency metrics, and compositionality.

## 3.2 Imitation Phase

Iterated learning is known to add compressibility pressure to emergent communication schemes [34], resulting in languages that are "easy to learn", thus favoring simple communication schemes that can be compressed: this is known as a *communication (or generational) bottleneck*. It is argued [34] that languages that remain relatively stable even when a learner only observes a small subset of the language of the previous generation need to be compositional.

Our case (and in fact, most deep learning applications of IL) differ from the original IL formulation [34], in that the transmission bottleneck does not constrict the amount of examples presented to the student, but rather constricts the information obtainable from these examples in order to perfectly reconstruct the language. This paradigm shift for transmission bottlenecks is at the heart of discrete bottlenecks [55, 58, 57] and noisy bottlenecks [56].

We next pass to describe our proposed imitation scheme and its motivations compared to previous methods.

### 3.2.1 Final-State Imitation (FiSI)

As usual in IL, we apply a teacher-student regime to initialize the next iteration sender $S_{\phi^{t+1}}$.

Our main proposed novelties for the imitation phase are as follows.

**State space imitation:** As opposed to other IL implementations [55, 55, 44], our models are not directed to match the student protocols from different iterations in the message space, but rather in the state space, using the frozen receiver network from the interaction phase. This change allows a wider range of sender strategies, as the new loss is more permissive.

**Final state reconstruction:** During imitation phase we use the following reconstruction loss, where $R_t^\star = R_{\omega^t}, S_t^\star = S_{\phi^t}$ are frozen receiver and sender from the previous iteration and $S_{t+1} = S_{\phi^{t+1}}$ is the newly trained sender:

$$\mathcal{L}_{\phi^{t+1}}^{rec} := \mathbb{E}_x \left[ d_{\mathcal{X}}(R_t^\star(S_t^\star(x)), R_t^\star(S_{\phi^{t+1}}(x))) \right]. \tag{2}$$

Our loss $\mathcal{L}_{\phi^{t+1}}^{rec}$ only depends on the state associated to the full message, without testing reconstruction on sub-messages. This change allows the intermediate message tokens of the student sender $S_{\phi^{t+1}}(x)_i, i < C$ to drift further away from the ones of the teacher mapping $S_t^\star(x)_i$ and thus more easily find different strategies with the same end result.

State space imitation is justified as follows. Note that as message space is discrete, there exists a value $\delta_0$ such that any two messages at distance $< \delta_0$ in fact coincide. Then $d_{\mathcal{M}}(S_t^\star(x), S_{t+1}(x)) < \delta_0$ always implies $d_{\mathcal{X}}(R_t^\star(S_t^\star(x)), R_t^\star(S_{t+1}(x))) = 0$, while the vice-versa is not always true: if $R_t^\star$ is not injective, any $S_{t+1}(x) \in (R_t^\star)^{-1}(S_t^\star(x))$ will preserve zero distance of $R_t^\star$-images. If $R_t^\star$ has mild Lipschitz regularity assumption, this principle extends to the regime of small loss, showing that our new loss is less restrictive on the choices of $S_{t+1}$ than the traditional message space loss. Thus, state space imitation enables wider sender strategy exploration without penalization losses.

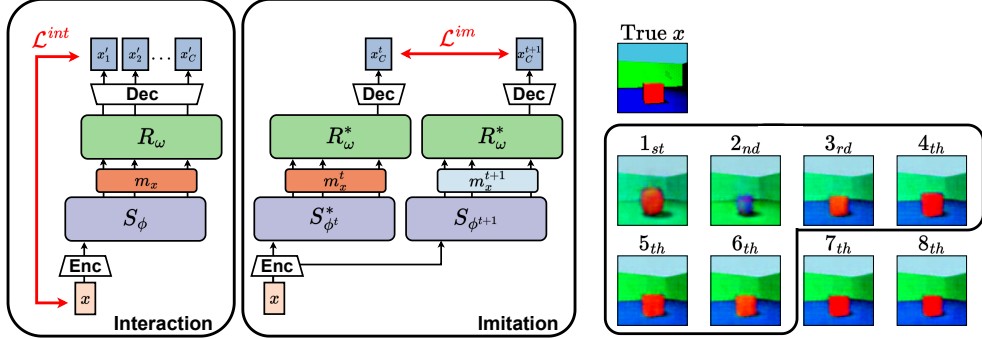

(a) Overview of our proposed architecture   (b) Qualitative example of image decoding

Figure 1: **(a) Overview of our proposed architecture. Interaction Phase:** The sender $S_{\phi^t}$ and receiver $R_{\omega^t}$ are jointly trained to minimize the reconstruction error between the input state $x$ and the predicted states $\{x'\}$, by encoding $x$ into a message $m_x^t$. **Imitation Phase:** A new sender $S_{\phi^{t+1}}$ is trained to imitate the final predicted output of $R_{\omega^t}(m_x^t)$, while also maximizing pairwise message diversity to encourage exploration. **(b) Qualitative example of image decoding:** the receiver reconstructs the input image from the sender's message for each sub-message, giving a series of reconstructions. The *useful length* of a message corresponds to the first reconstruction that has error below a threshold $\epsilon > 0$: in this example the last two message tokens are not useful in that they do not add to the reconstruction accuracy.

The choice of using just final states is justified because final state reconstruction allows greater freedom during imitation for the student sender. By only constraining the reconstruction of the full message, we allow much more freedom for the encoding strategy of $S(x)$ at earlier sub-messages at each iteration (see Proposition E.1 for a formal statement and proof).

### 3.2.2 Pairwise Distance Maximization (PDM)

To further improve protocol diversity and promote exploration of the language space during the imitation phase, we introduce a regularization objective that promotes dissimilarity between messages within each batch. In addition to enabling a larger search space for optimal student strategies. Enhancing the diversity of student strategies also fits the general IL argument according to which compositional languages are the most resistant to noisy language transmission [34, 37, 36].

Specifically, we approximate the Hamming distance by computing the position-wise cosine dissimilarity between the probability distributions over symbols in each message.[3] This encourages the sender to generate messages that are maximally distinct while remaining semantically aligned with the teacher protocol. The strength of this regularization is controlled by a hyperparameter $\beta$.

The resulting imitation loss is defined as:

$$\mathcal{L}_{\phi^{t+1}}^{im} := \mathcal{L}_{\phi^{t+1}}^{rec} + \beta \frac{1}{N_{batch}^2} \sum_{i,j} d(S_{\phi^{t+1}}(x_i), S_{\phi^{t+1}}(x_j)), \tag{3}$$

where $d(\cdot, \cdot)$ is the mean cosine similarity between corresponding symbol distributions.

In § F.4, we show that this regularization term provides a lower bound on entropy maximization objectives [69], and serves as an upper bound for contrastive losses such as NT-Xent [11], thus connecting our formulation to well-established principles in representation learning.

## 4 Experimental setup

### 4.1 Datasets

**Shapes3D**  We tested our framework using the Shapes3D [50] dataset, which consists of colored images of 3D geometric shapes, with 6 underlying generating factors $G$: floor hue, wall hue, object hue, shape, scale, and orientation. The total amount of attribute-value combinations is $480,000$.

**MPI3D**  We also evaluated using the compositional split of the MPI3D dataset [23]. This dataset consists of colored images of a robot arm interacting with objects, rendered in controlled 3D scenes with 7 underlying generative factors $G$: object color, object shape, object size, camera height, background color, horizontal arm position, and vertical arm position. The dataset comprises $1,036,800$ unique combinations of these factor values.

For both datasets we use the compositional split from Schott et al. [62], which ensures all attribute values appear in training, yet some combinations are reserved for the testing set (See §D for a proof that this situation would anyways hold for random train/test splittings with high probability).

### 4.2 Evaluation Metrics

Following Chaabouni et al. [9], we assess compositionality using *Topographic Similarity* (TopSim) [5], a widely used proxy for compositionality. TopSim measures the correlation between pairwise Hamming distances in message space and the generating factor space, capturing the extent to which semantically similar inputs yield similar messages under a structured encoding.

To evaluate communication efficiency, we define the *useful length* $\hat{\ell}_\epsilon(x)$, which estimates the minimal prefix of a message necessary to achieve near-maximal reconstruction quality. Formally:

$$\hat{\ell}_\epsilon(x) := \min \left\{ i \in \{1, \ldots, C\} : \text{MSE}\left(x, R_\omega(S_\phi(x)_{[i]})\right) \leq \epsilon \right\},$$

where $C$ is the maximum message length and $\ell_\epsilon(x) = C$ if the threshold is unmet.

We choose $\epsilon$ by viewing the loss distributions on both tested datasets and estimating a common plateau point for each. The full position-wise loss values can be found in the supplementary material.

Finally, to approximate the expressivity of the communication protocol, we evaluate reconstruction quality using the MSE between the generated image and the original input.

We discuss the use of additional language metrics in §I.

## 5 Results

For all experiments, we report the mean and standard error of 10 random seeds. To assess the robustness of our findings, we also perform permutation tests to evaluate the statistical significance of

---

[3]For one-hot encoded messages, this formulation is equivalent to maximizing the Hamming distance.

Table 1: Performance comparison across all experiments on the SHAPES3D and MPI3D datasets. We report Topographic Similarity (TopSim ↑), Useful Length ($\hat{\ell}_\epsilon$ ↓), and Last Symbol MSE (↓) for each ablation. The results are grouped by the components of our proposed framework: **Interaction**, **Imitation**, and **Regularization**. The synergetic mixture of our proposed methods (PD+FiSI+PDM) consistently improves compositionality and message efficiency over the baseline and prior variants.

| | | TopSim ↑ | $\hat{\ell}_{2\times10^{-1}}$ ↓ | Last symbol MSE ↓ |
|---|---|---|---|---|
| **Interaction** | Full message reconstruction, no IL (Baseline) | $0.244 \pm 0.002$ | $10.0 \pm 0.0$ | $0.212 \pm 0.003$ |
| | Progressive Decoding(PD, ours) | $0.270 \pm 0.001$ | $7.5 \pm 0.215$ | $0.238 \pm 0.007$ |
| **Imitation** | Message imitation (baseline) | $0.257 \pm 0.005$ | $9.9 \pm 0.3$ | $0.200 \pm 0.007$ |
| | Full state imitation | $0.256 \pm 0.003$ | $9.9 \pm 0.3$ | $0.179 \pm 0.003$ |
| | Final-State Imitation (FiSI, ours) | $0.283 \pm 0.003$ | $10.0 \pm 0.0$ | $\mathbf{0.176 \pm 0.002}$ |
| **Regularization** | PD+FiSI+KoLeo $\lambda = 1.5$ | $0.256 \pm 0.002$ | $8.555 \pm 0.167$ | $0.179 \pm 0.003$ |
| | PD+FiSI+PDM (ours) $\lambda = 1.5$ | $\mathbf{0.292 \pm 0.002}$ | $\mathbf{7.0 \pm 0.155}$ | $0.194 \pm 0.005$ |

(a) Shapes3D

| | | TopSim ↑ | $\hat{\ell}_{1.95\times10^{-2}}$ ↓ | Last symbol MSE ↓ |
|---|---|---|---|---|
| **Interaction** | Full message reconstruction, no IL (Baseline) | $0.133 \pm 0.001$ | $9.3 \pm 0.002$ | $\mathbf{0.015 \pm 0.0}$ |
| | Progressive Decoding(PD, ours) | $0.137 \pm 0.001$ | $\mathbf{6.6 \pm 0.341}$ | $\mathbf{0.015 \pm 0.0}$ |
| **Imitation** | Message imitation (baseline) | $0.135 \pm 0.001$ | $9.733 \pm 0.029$ | $0.016 \pm 0.0$ |
| | Full state imitation | $0.137 \pm 0.001$ | $9.923 \pm 0.020$ | $0.018 \pm 0.0$ |
| | Final-State Imitation (FiSI, ours) | $\mathbf{0.156 \pm 0.001}$ | $9.7 \pm 0.046$ | $0.02 \pm 0.0$ |
| **Regularization** | PD+FiSI+KoLeo $\lambda = 1.5$ | $0.147 \pm 0.002$ | $8.9 \pm 0.221$ | $0.02 \pm 0.0$ |
| | PD+FiSI+PDM (ours) $\lambda = 1.5$ | $0.153 \pm 0.001$ | $9.0 \pm 0.167$ | $0.02 \pm 0.0$ |

(b) MPI3D

the observed differences (see §J.1). Additionally, we include qualitative reconstruction experiments in §H, where we visually compare the outputs of all evaluated methods across progressive decoding steps. Further implementation details are provided in §B.

## 5.1 Progressive Decoding reduces the message's useful length

We begin by evaluating the effect of incorporating PD during the interactive phase of training. As shown in the first two rows of Table 1, this modification leads to a 25-29% reduction in the useful message length, indicating more efficient communication. These gains are achieved without compromising expressivity on MPI3D and with only a slight compromise on Shapes3D, as indicated by stable reconstruction quality in the former and a minor degradation in the latter.

To further promote efficient and structured communication, we introduce a geometric penalty term $\lambda$ that penalizes reconstruction error more heavily when longer messages are used. As shown in Figure 2, increasing $\lambda$ up to values near 1.5 consistently improves reconstruction quality and useful length, indicating higher expressivity and efficiency. However, when $\lambda$ becomes too large, the reconstruction error incurred at the end of the message dominates the loss function, effectively nullifying PD. We found that setting $\lambda = 1.5$ yields favorable results in efficiency and expressivity, while maintaining high compressibility.

## 5.2 Final State Imitation increases TopSim

Second, we investigate the impact of final state imitation on the structure of the emergent language.

In these experiments, we compare our approach to a baseline model without imitation and to a standard IL variant that imitates the teacher's message directly—a strategy commonly adopted in prior work for optimizing this phase [55, 56, 44].

As shown in the imitation section in Table 1, in the MPI3D dataset, we find that our *final state imitation* method consistently enhances the compositionality of the emergent messages. It outperforms the non-IL baseline and surpasses message-level imitation significantly (§ J.1).

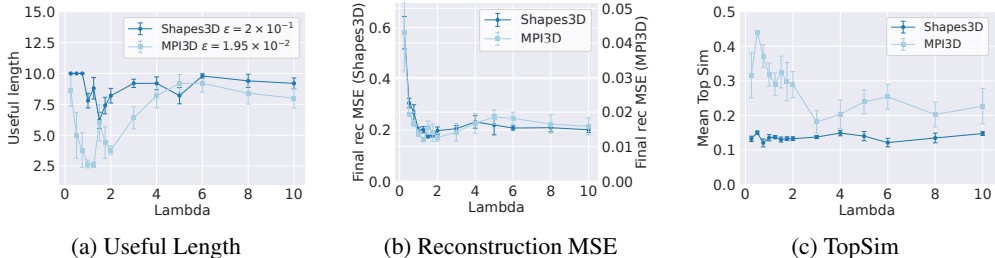

|  | (a) Useful Length | (b) Reconstruction MSE | (c) TopSim |
|--|--|--|--|

Figure 2: **Effect of geometric penalty** $\lambda$ **on emergent communication.** We introduce a geometric weighting term $\lambda$ that penalizes reconstruction error more strongly for longer messages. Increasing $\lambda$ initially improves useful length (a). However, once a certain threshold is met, useful length and reconstruction error (b) increase. Values close to 1.5 of $\lambda$ strike a favorable balance—achieving efficient and expressive communication.

More broadly, our results support prior findings that IL promotes greater structure in emergent communication protocols [44, 55]. They also underscore the value of shifting the learning objective from message imitation to image reconstruction, which enforces a tighter generational bottleneck by requiring the student to recover the intended output without access to the original message.

Moreover, this setup increases flexibility in protocol exploration by constraining the student only to the reconstruction target, not the exact message form. Permitting generational variation in message structure while preserving semantic fidelity, promoting diversity without sacrificing expressivity.

Together, these factors exert pressure toward discovering more systematic and compositional protocols—structures that are inherently more transmissible and robust across generations.

### 5.3 Pairwise Distance Maximization increases compositionality and efficiency

We evaluate a model that combines all previously identified best-performing components—specifically, $\lambda = 1.5$, final-state imitation, PD and further introduce an entropy-based regularization term using PDM. We compare this with the KoLeo entropy estimator proposed by Sablayrolles et al. [61].

When examining the effect of incorporating PDM, Table 1 shows that PDM outperforms both FiSi and PD in fostering structure in the message. On the Shapes3D dataset, PDM leads to a marked improvement in TopSim and a lower useful message length, indicating more efficient and structured communication, albeit with a slight increase in reconstruction error. For MPI3D, this configuration doesn't achieve the highest TopSim, albeit close to the best, and it increases useful length and leads to a marginally higher reconstruction loss compared to using only PD. This highlights a trade-off between expressivity and compressibility, in line with prior theoretical accounts on communication efficiency and compositionality [53, 34, 49].

When comparing PDM to the KoLeo estimator, Regularization section in Table 1, we find that on Shapes3D, PDM achieves superior TopSim and shorter useful length, with slight reconstruction gains. On MPI3D, KoLeo leads to a lower useful length but exhibits reduced compositionality, with both regularizers showing similar reconstruction performance.

### 5.4 Comparison to disentangled representation learning

Following the comparison made by Xu et al. [71], we test our generated messages against standard self-supervised disentangled representation learning frameworks, namely $\beta$-VAE and $\beta$-TCVAE, as well as the baseline VAE+EL [71]. For the continuous models, we extract the first half of the latent vector (corresponding to the predicted means $\mu$) and evaluate it using the DCI disentanglement score [18]. In addition, we train a two-layer MLP to predict ground-truth generative factors, reporting RMSE for continuous factors and classification accuracy for categorical ones.

For the discrete models, we use the predicted messages for each image. We calculate the DCI disentanglement as-is, and use an embedding matrix to transform the discrete messages into the same dimension as the continuous vectors for the MLP. Training is conducted on three subsets of size 1000

Table 2: Disentanglement metrics for the Shapes3D and MPI3D datasets. We report the DCI score to evaluate disentanglement quality, and the RMSE and classification accuracy of a linear probe trained on the latent representations (both message and vector embeddings) to assess their suitability for downstream tasks.

| Dataset | Model | Disentanglement ↑ | RMSE ↓ | Acc ↑ |
|---------|-------|-------------------|--------|-------|
| Shapes3D | $\beta$-VAE | $0.045 \pm 0.004$ | $2.460 \pm 0.144$ | $0.702 \pm 0.025$ |
| | $\beta$-TCVAE | $0.043 \pm 0.005$ | $\mathbf{2.378 \pm 0.123}$ | $\mathbf{0.721 \pm 0.028}$ |
| | VAE + EL | $0.108 \pm 0.000$ | $3.168 \pm 0.027$ | $0.343 \pm 0.019$ |
| | VAE + CELEBI | $\mathbf{0.112 \pm 0.001}$ | $2.932 \pm 0.045$ | $0.453 \pm 0.024$ |
| MPI3D | $\beta$-VAE | $0.031 \pm 0.001$ | $21.819 \pm 0.111$ | $0.581 \pm 0.006$ |
| | $\beta$-TCVAE | $0.031 \pm 0.001$ | $21.854 \pm 0.188$ | $\mathbf{0.582 \pm 0.005}$ |
| | VAE + EL | $0.114 \pm 0.002$ | $15.971 \pm 0.301$ | $0.529 \pm 0.006$ |
| | VAE + CELEBI | $\mathbf{0.137 \pm 0.002}$ | $\mathbf{14.82 \pm 0.012}$ | $0.542 \pm 0.002$ |

of the training set and averaged. Evaluations are conducted on the test sets of the compositional splits for the Shapes3D and MPI3D datasets.

Across both datasets, our method (VAE+CELEBI) demonstrates consistent improvements in disentanglement over baseline methods. On Shapes3D, we observe a substantial increase in DCI scores compared to continuous models and the discrete baseline (see statistical significance tests in § J.2). However, both discrete models fall behind the continuous baselines in downstream accuracy and RMSE, suggesting a trade-off between representation interpretability and usefulness in downstream tasks, which is consistent with previous work indicating that discrete representations may be less accessible to simple classifiers (as shown in [72]), and that inductive biases for compositionality do not imply disentanglement [51].

On MPI3D, our method slightly under-performs when compared to the continuous baselines in categorical accuracy, however greatly outperforming all baselines in continuous regression. We speculate that this discrepancy arises from the lower correlation between pixel-level input statistics and continuous generative factors in MPI3D as opposed to Shapes3D, which may favor communication-based models over purely reconstructive.

## 6 Related Work

The use of language reference games (LRGs) in conjunction with iterated learning for the purpose of learning self-supervised representations remains underexplored. Xu et al. [71] compare an emergent language autoencoder—comprising a visual backbone and an LSTM-based sender and receiver—with disentanglement frameworks such as the $\beta$-VAE and $\beta$-TCVAE. They find that the representations induced by the emergent language model generalize better in downstream tasks, highlighting the potential of symbolic communication as a compositional bottleneck.

Many previous works have explored using emergent language to induce compositional behavior [31, 17, 2, 16, 8]. Several works utilize a referential LSG in conjunction with IL, using predicted messages as a generational bottleneck [55, 44]. Alternative bottlenecks for emergent language have been proposed, including simplicial embeddings [56, 19], code-books [74] and noisy channels [65]. Our message tokens can also be considered as roughly comparable to the slots in slot-attention architectures such as [22, 24, 46, 27, 45, 21, 48, 4, 1], however note that among other differences, the slot-attention typically additive decoding layer [7, 38] does not compare directly to our proposed Progressive Decoding.

To replicate the Zipfian distributions observed in natural languages [76, 29], Rita et al. [57] propose the LazImpa framework. The LazImpa *Impatient listener* loss is similar to our proposed Progressive Decoding, in that they both aim to induce incremental informativeness and efficiency by reconstructing the input at every timestep. However LazImpa is optimized for a referential game task (and not a reconstruction task as ours) and uses a linear length penalization in conjunction with a cross-entropy loss. Our proposal instead uses the actual reconstruction loss and the length penalization is exponential and based on a tunable parameter $\lambda$.

# 7 Limitations and Future Work

Our experiments are conducted exclusively on the synthetic datasets *Shapes3D* and *MPI3D* [6, 23], which feature clean, disentangled generative factors and lack observational noise. In future work, we plan to extend our the same principle of making explicit pressures towards complexity reduction of underlying representations to natural image datasets where factor supervision may be unavailable or weakly defined. Furthermore, we aim to formalize the notion of language compression defined in IL in terms of Kolmogorov complexity, for representing datasets with more complex compositional structures.

Moreover, it is debatable whether MSE is the most appropriate choice for the self-supervised loss function, especially in the context of natural image data. While MSE is a simple choice, future work may explore reconstruction losses that better align with human visual perception, such as perceptual similarity metrics based on deep feature embeddings (e.g., LPIPS [73]), or adversarial and contrastive losses that promote more semantically structured representations.

Additionally, we froze the visual backbone across all EL model variants trained on the same dataset. While this ensures controlled comparisons within each dataset, it may introduce an initialization bias when comparing with continuous baselines, where the encoder is jointly optimized. Although our results remain statistically significant, future work could investigate whether co-training the vision encoder yields improved alignment between symbolic and perceptual representations, or leads to new trade-offs in compression and expressivity.

Finally, our experiments were restricted to compositional generalization metrics and reconstruction-based evaluation. Other facets of emergent communication, such as robustness and interpretability, are left for future work. We believe our framework offers a strong foundation for such extensions and a tractable setting to explore the interplay between compression, efficiency, and generalization.

# 8 Conclusion

We introduce the CELEBI framework with three novel mechanisms for enhancing compositional learning in IL frameworks, as well as mathematical justification for their design. First, with *Progressive Decoding* (PD) we reward informative communication at each step of the message, creating an inductive bias toward efficient and distributed representations. We show that PD promotes both message compressibility and compositional alignment.

Second, in our *Final State Imitation* (FiSI) in IL, the student is trained to reproduce the final prediction of the teacher, rather than the message itself. This shift enables greater exploration of the message space while preserving semantic consistency. We provided theoretical motivation and empirical evidence that this approach yields more expressive and compositional protocols, outperforming traditional message imitation in TopSim and reconstruction metrics.

Third, our regularizer based on *Pairwise Distance Maximization* (PDM), which provably approximates entropy maximization over messages, serves as a practical inductive bias for promoting diversity and structure in emergent languages, particularly during the imitation phase.

Together, these contributions enrich the IL framework, and give new insights into how training dynamics and inductive pressures shape the emergence of language-like representations. As confirmed by our empirical findings, the proposed methods lead to more compositional and generalizable communication schemes.

## Acknowledgments and Disclosure of Funding

This work was supported by ANID Chile, Fondecyt Regular grant 1231724, as well research centers of excellence with code FB210017 (Basal CENIA), ICN2021_004 (Millenium iHealth), and ICN17_002 (Millenium IMFD).

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

# A  Background

## A.1  Iterated Learning

The IL model of cultural evolution [34] proposes to emulate the emergence of language through the interactions of adult and new learning agents. In the original formulation, a space of pairs of signal + meaning is randomly generated, and in each iteration, a learning agent is partially exposed to the language learned by the adult agent in the previous iteration. The learning agent becomes the adult agent for the next iteration, and transmits a subset of its learned "language". By iterating this process, the initial random language gains structure and observable properties, such as compositionality.

This framework has been widely extended to include an additional task [44, 56, 75], instead of simply reshaping the language. Under this formulation, the training regime can be separated into two parts: Interaction (**Interaction**), where an agent is optimized to solve the additional task, and Imitation (**Imitation**), where a newly initialized agent "learns" from the previous agent through a shared learned language.

## A.2  Lewis reconstruction game

The Lewis reconstruction game (LRG) [58] is a special case of the Lewis Signaling Game (LSG) [43] framework, in which two agents cooperate to reconstruct an observed object. Specifically, a sender agent parameterized by $\varphi$ observes an object $x$ from an object space $\mathcal{X}$ and produces a message $m \in \mathcal{M}$, where $\mathcal{M} = V^\star$ is the set of all possible strings formed by concatenating symbols drawn from a finite vocabulary $V$. A receiver agent parameterized by $\omega$ observes $m$ and produces a reconstruction $x'$. Both models are optimized to minimize a reconstruction loss $\mathcal{L}_R(x, x')$.

Importantly, in the original LSG, the receiver is tasked with predicting $x$ from a finite set of states, whereas in this formulation, the problem becomes a regression task, and $\mathcal{X}$ can be a continuous space.

# B  Implementation Details & Hyperparameters

## B.1  Baseline Model

- **Sender and Receiver:** Both consist of a single-layer LSTM with an embedding size of 64 and a hidden size of 256. Outputs are passed through a two-layer MLP with ReLU activations.
- **Communication Channel:** The vocabulary size is set to $|V| = 15$ and messages are composed of $C = 10$ discrete symbols. These values are chosen such that the amount of posible messages ($5.76 \times 10^12$) greatly outnumbers the amount of states. The Gumbel-Softmax bottleneck is used without straight-through estimation.
- **VAE Backbone:** The VAE is implemented using the disentanglement_lib [47] python library, using the default setting of two convolutional layers of 32 filters, with kernel size 4 and stride 2, two convolutional layers of 64 filters, with kernel size 2 and stride 2, each with ReLU activation, as well as a linear layer of size 256 and ReLU activation and an output layer with size 2*128 for concatenated $\mu$s and log$\sigma$s. The encoder is mirrored replacing convolutional layers with transposed convolutions, and an input layer of size 128 and ReLU activations on all layers save the output.
- **VAE Training:** The VAE is pretrained for 15 epochs using the Adam optimizer with a learning rate of $1 \times 10^{-3}$.

The interaction-imitation games were run for a maximum of 100 iterations, alternating one full epoch in each phase per iteration.

We used early stopping based on the validation MSE of the final reconstruction, starting from epoch 5, with a minimum $\delta$ of $1 \times 10^{-3}$ and patience of 5. We used the Adam optimizer with default parameters and learning rates of $1 \times 10^{-3}$ and $1 \times 10^{-4}$ for Shapes3D [6] and MPI3D [23], respectively, as we found the training for the latter became unstable at higher learning rates.

We found that **Interaction** required smaller batch sizes to converge, whereas the **Imitation** phase **PDM** becomes a better entropy approximation at higher batch sizes, however scaling quadratically

in computation. Therefore, we set the **Interaction** and **Imitation** batch sizes to 256 and 512, respectively.

To implement **Imitation** we defined a receiver $R_\omega$ and two senders $S_{\phi_1}$ and $S_{\phi_2}$ with their own optimizers. After participating in an **Interaction** iteration, $\phi^1$ and $\omega$ are frozen and used to train $S_{\phi_2}$ as described in §3.2. After **Imitation**, the weights $\phi_2$ are copied to $\phi_1$, and $\phi_2$ is reset. Importantly, the optimizers for $\phi_1$ and $\phi_2$ are **never reset nor copied between them.**

## C Compute Resources

All experiments were conducted on our internal laboratory cluster, using NVIDIA A40 GPUs with 48 GB of VRAM. Each job was allocated a single GPU with 8 GB of VRAM usage on average, alongside access to a CPU with 20 cores, 118 GB of RAM, and local SSD storage for datasets and model checkpoints.

The experiments reported in this paper comprise 360 training runs, totaling approximately 180 GPU hours. Individual runs varied in length from 15 to 50 minutes, depending on early stopping criteria. These runs represent the finalized experiments whose results are presented in the main text.

In total, the research project required 2,310 experimental runs over the course of development, amounting to approximately 245 GPU days (5,880 GPU hours) of compute time. This includes preliminary experiments, ablations, and failed runs that were instrumental to model and protocol design but are not individually reported in the paper.

### C.1 Useful length $\epsilon$

To define the useful length $\epsilon$ for both datasets, we observed reconstruction loss distribution over all positions in the message (see 3) for different $\lambda$ values, and estimated an average plateau point. For MPI3D we used a more conservative and precise threshold, as the reconstruction error varied more between methods.

Additionally, we analyzed the useful length graph for multiple $\epsilon$ values, and discarded those values where most models converged to either maximum or minimum possible length (see 4).

## D Recovering compositional structure from partially observed data

For the sake of clarity of treatment, we assume here that $\mathbf{G} = [G_1, \ldots, G_n]$ in whichi factor $G_i$ has the same number $N$ of possible values, and as a consequence $\mathcal{G} = [1:N]^n$. Our results will extend directly to the general $\mathcal{G}$ case as shown in Rmk. D.4.

We work under the simplified deterministic generation hypotheses, in which our dataset is $\mathcal{D} = \mathsf{GenX}(\mathcal{G}) \subset \mathcal{X}$ in which $\mathsf{GenX}$ is a deterministic injective function.

Our main result is the following:

**Theorem D.1.** *Assume that $\mathcal{G} = [1:N]^n$ and that $\mathsf{GenX} : \mathcal{G} \to \mathcal{X}$ is an injective function, and that $\mathcal{D}_{train} \subseteq \mathcal{D}$ has cardinality $|\mathcal{D}_{train}| = p|\mathcal{D}|$ for some $p \in (0,1)$ such that $pN^n$ is an integer.*

*We assume that the compositional structure of $\mathcal{G}$ is such that the following holds:*

*(A) For some $k < n$, if we observe a set of data whose generating factor combinations feature all possible combinations of generating factor values $(G_{i_1}, \ldots, G_{i_k})$ for all choices of indices $1 \leq i_1 < \cdots < i_k \leq n$ then this is sufficient to reconstruct $\mathcal{G}$ to good accuracy.*

*Then as $|\mathcal{G}| \to \infty$ the probability that $\mathcal{D}_{train}$ is sufficient to reconstruct $\mathcal{G}$ tends to 1.*

Here **(A)** makes rigorous the common assumption that having a training set that features examples for diverse enough combinations of factors is sufficient for reconstructing all combinatorial generators **G**.

Note that in this work we **do not discuss**

1. what features of the pre-trained encodings $\mathcal{G} \to \mathcal{D}$ and what requirements on the compositional structure of $\mathcal{G}$ would allow to actually guarantee this assumption,

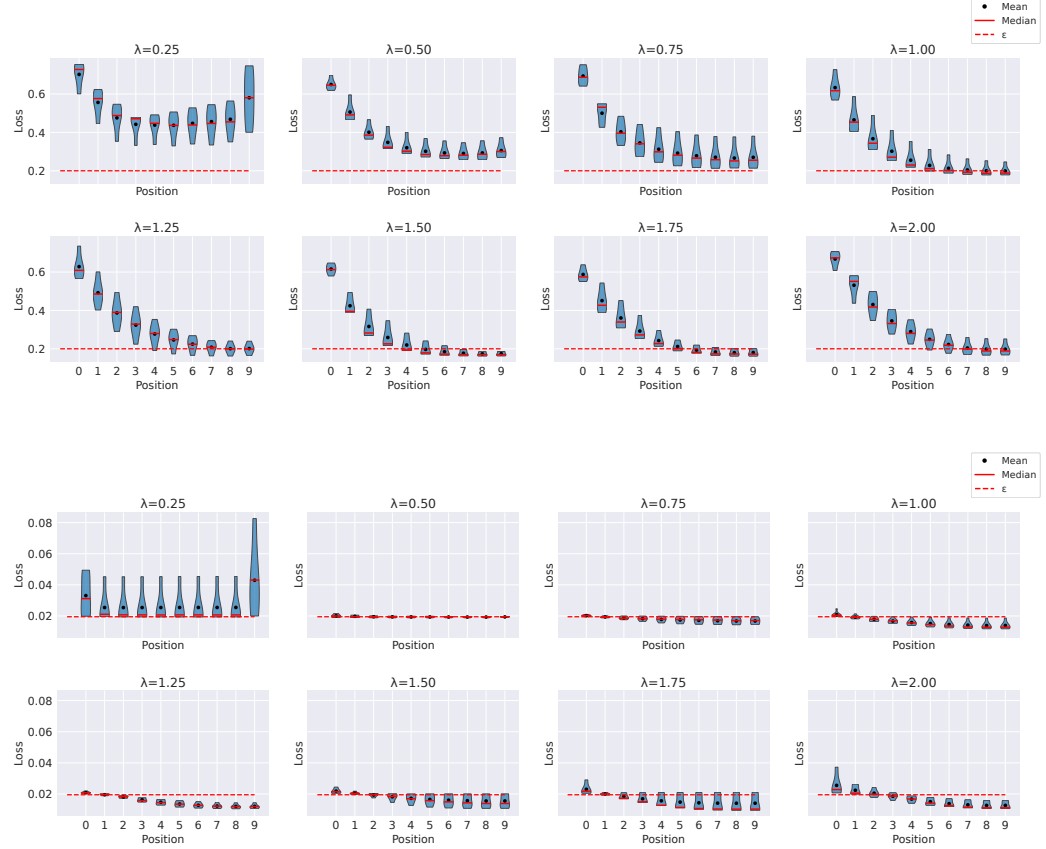

Figure 3: Loss distribution over positions and $\lambda$ values for **(a)** Shapes3D and **(b)** MPI3D

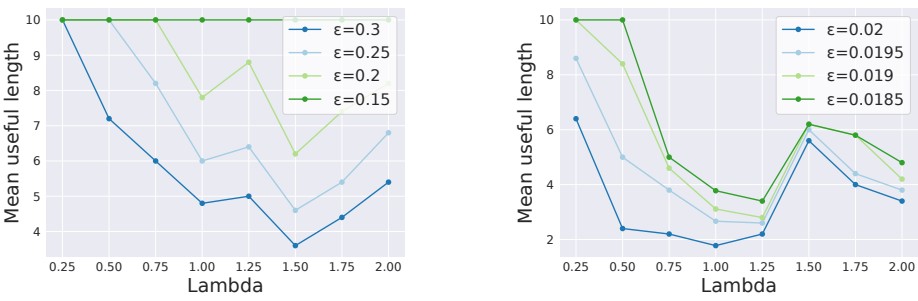

Figure 4: Useful length values for different $\epsilon$ and $\lambda$ values for **(a)** Shapes3D and **(b)** MPI3D

    2. alternatives for a rigorous definition of what is meant by "to good accuracy" in the statement of assumption **(A)**.

Once the above points are settled and defined, we can replace **(A)** by an explicit requirement. Since however these points seem like an ambitious whole research line and are far beyond the scope of this work, we leave the definition of "to good accuracy" and the proof of reconstruction to future works.

Thus, **introducing (A) allows us to prove fully rigorous statements such as Thm. D.3, in absence of a full theory of compositional reconstruction**.

Nevertheless, we make the following remarks:

- There is a natural trade-off between assumptions on how big $\mathcal{D}_{train}$ should be, vs. how complicated the compositional reconstruction should be allowed to be for reconstruction to be feasible: the richer set of such combinations we require $\mathcal{D}_{train}$ to contain (thus making more restrictive assumptions on $\mathcal{D}_{train}$), the lower is the requirement for our encoding framework and for the compositional structure for **(A)** to hold.

- It seems natural to conjecture that if assumption **(A)** does not hold for large value of $k$, then there is no hope for the compositional reconstruction to be achievable. Thus lowering the requirement on "good enough accuracy" to make **(A)** hold seems the only avenue of future research.

Theorem D.1 is a consequence of below Theorem D.3 that proves that assumption **(A)** holds with probability tending to 1 as $|\mathcal{G}| \to \infty$ and $|\mathcal{D}_{train}|/|\mathcal{G}| = p$ is bounded away from zero. What Theorem D.3 says is that for large sets of generators, it will become overwhelmingly unlikely that $\mathcal{D}_{train}$ does not contain enough data to reconstruct the generators.

### D.1 Observing $\mathcal{D}_{train}$ allows to observe finite statistics of the factors with high probability

Our first result for this section is that when $\mathcal{D}_{train} \subseteq \mathcal{D}$ is chosen uniformly at random amongst sets of size $p|\mathcal{D}|$, for $|\mathcal{D}| = |\mathcal{G}|$ large enough, we find that with overwhelming probability all possible values of each of the factors $G_1, \ldots, G_n$ are achieved by some elements of $\mathcal{D}_{train}$. We state the result in Proposition D.2, whose proof in our view contains the main ideas useful also for the full result of Theorem D.3, which can be considered our main technical result for this section.

The first result is as follows:

**Proposition D.2.** *Assume that for $n, N \geq 2$ we have $\mathcal{G} = \{\mathbf{G} = [G_1, \ldots, G_n] : (\forall i \leq n)1 \leq G_i \leq N\} = [1 : N]^n$ and that $\mathsf{GenX} : \mathcal{G} \to \mathcal{X}$ is an injective function with image $\mathcal{D}$ as above. Denote by $\mathcal{G}_{train} := \mathsf{GenX}^{-1}(\mathcal{D}_{train})$ the generating factors related to the training set, and assume that $\mathcal{D}_{train} \subset \mathcal{D}$ is chosen uniformly at random amongst subsets of cardinality $p|\mathcal{D}|$, for some $p \in (0,1)$ such that $pN^n$ is an integer. Then*

$$\mathbb{P}\{(\exists i \leq n)(\exists j \leq N)(\forall \mathbf{G} = [G_1, \ldots, G_n] \in \mathcal{G}_{train})\, G_i \neq j\} \leq nN\left(1 - \frac{1}{N}\right)^{pN^n}, \quad (4)$$

*in particular at fixed $p$, the above event has probability tending to zero as $N^n \to \infty$.*

Thus, in an equivalent formulation of the last part of the above proposition, we have that in the limit of large $|\mathcal{G}| = N^n$, the probability that each generating factor $G_i$ takes all of its possible values on some generated element of $\mathcal{G}_{train}$ tends to 1.

*Proof.* The possible choices of $\mathcal{G}_{train} \subseteq \mathcal{G}$ of cardinality $p|\mathcal{G}| = pN^n$ are $\frac{(N^n)!}{(pN^n)!((1-p)N^n)!}$. Then, calling $A_{ij}$ the set of elements $\mathbf{G} \in \mathcal{G}_{train}$ such that $G_i \neq j$, we find that $|A_{ij}| = N^{n-1}(N-1) = N^n \frac{N-1}{N}$, so that the number of possible choices of $\mathcal{G}_{train} \subseteq A_{ij}$ of cardinality $pN^n$ is given by $\frac{(N^n \frac{N-1}{N})!}{(pN^n)!(N^n(\frac{N-1}{N}-p))!}$. Now we bound the probability of the event on the left hand side in (4) via a union bound over the $Nn$ possible choices of $A_{ij}$ for $1 \leq i \leq n, 1 \leq j \leq N$, so that

$$
\mathbb{P}\left\{ \begin{array}{l} \exists i \leq n,\ \exists j \leq N, \\ \forall \mathbf{G} = [G_1, \ldots, G_n] \in \mathcal{G}_{train}, \\ G_i \neq j \end{array} \right\} \begin{array}{l} \leq \quad nN\dfrac{(pN^n)!((1-p)N^n)!}{(N^n)!}\dfrac{(N^n \frac{N-1}{N})!}{(pN^n)!(N^n(\frac{N-1}{N}-p))!} \\[4mm] = \quad nN\dfrac{N^n \frac{N-1}{N}}{N^n}\dfrac{N^n \frac{N-1}{N}-1}{N^n-1}\cdots\dfrac{N^n \frac{N-1}{N}-pN^n+1}{N^n-pN^n+1} \\[4mm] \leq \quad nN\left(1 - \dfrac{1}{N}\right)^{pN^n}, \end{array}
$$

in which in the first line we used a union bound together with the computations of combinatorial coefficients from above, in the second line we performed a simplification and reordering, and in the

third line we used the fact that $1 - 1/N = (N^n(N-1)/N)/N^n > (N^n(N-1)/N - i)/(N^n - i)$ for all $1 \leq i < pN^n - 1$.

Finally, considering the limit $N^n \to \infty$ of the upper bound in (4), we have two cases: (1) if $N \geq 2$ stays bounded and $n \to \infty$, then we can bound $nN(1 - 1/N)^{pN^n} \leq C_1 n C_2^{C_3^n}$ with constants $C_2 \in (0, 1)$ and $C_3 > 1$, and by taking logarithms we check that the quantity tends to zero; (2) if $N \to \infty$ and $n \geq 2$, and in this case $(1 - 1/N)^{pN^n} \sim e^{-pN^{n-1}}$ and again by taking logarithms we find that the quantity tends to zero. □

In fact Prop. D.2 is gives us guarantees that it will be possible to observe all options for each $G_i$ in the training set, however this may not be sufficient for the main aim of recovering the compositional structure of $\mathcal{G}$. However we next prove, in an extension of Prop. D.2, that actually we will also observe with high probability **any possible combination of finitely many factors**:

**Theorem D.3.** *Under the same hypotheses on $\mathcal{G}$, GenX, $\mathcal{G}_{train}$, $p$ as in Prop. D.2, for any $k \leq n$ we have the following.*

$$\mathbb{P} \left\{ \begin{array}{ll} \exists i_1 < \cdots < i_k \leq n \\ \exists j_1, \ldots, j_k \leq N & (G_{i_1}, \ldots, G_{i_k}) \neq (j_1, \ldots, j_k) \\ \forall \mathbf{G} = [G_1, \ldots, G_n] \in \mathcal{G}_{train} \end{array} \right\}$$

$$\leq \frac{n!}{k!(n-k)!} N^k \left( 1 - \frac{1}{N^k} \right)^{pN^n}, \tag{5}$$

*which for fixed $p, k$ tends to zero as $N^n \to \infty$.*

*Proof.* As mentioned before, roughly the overall proof strategy is the same as for Prop. D.2. We introduce the sets $A_{i_1, \ldots, i_k}^{j_1, \ldots, j_k} = A_{\mathbf{i}}^{\mathbf{j}}$ and observe that the number of choices of $i_1 < \cdots < i_k \leq n$ is $\frac{n!}{k!(n-k)!}$ while the number of choices of $j_1, \ldots, j_k \leq N$ is $N^k$. Thus the number of sets of the form $A_{\mathbf{i}}^{\mathbf{j}}$ is $N_A := \frac{N^k n!}{k!(n-k)!}$. Furthermore, each set $A_{\mathbf{i}}^{\mathbf{j}}$ only imposes the constraint of entries of indices $i_1, \ldots, i_k$ from a $\mathbb{G}$ to avoid one combination of values $\vec{j}$, and thus it has cardinality $|A_{\mathbf{i}}^{\mathbf{j}}| = N^{n-k}(N^k - 1)$. Then the number of choices of $\mathcal{G}_{train} \subseteq A_{\mathbf{i}}^{\mathbf{j}}$ of cardinality $pN^n$ is just the combinatorial coefficient $\frac{(N^{n-k}(N^k - 1))!}{(pN^n)!(N^{n-k}(N^k - 1 - pN^k))!}$ and by union bound we then find by the same reasoning as in Prop. D.2,

$$
\begin{aligned}
\text{(left-hand side of (5))} &\leq N_A \frac{(pN^n)!((1-p)N^n)!}{(N^n)!} \frac{(N^{n-k}(N^k - 1))!}{(pN^n)!(N^{n-k}(N^k - 1 - pN^k))!} \\
&= N_A \frac{N^n - N^{n-k}}{N^n} \frac{N^n - N^{n-k} - 1}{N^n - 1} \cdots \frac{N^n - N^{n-k} - pN^n + 1}{N^n - pN^n + 1} \\
&\leq N_A \left( 1 - \frac{1}{N^k} \right)^{pN^n} = \frac{n!}{k!(n-k)!} N^k \left( 1 - \frac{1}{N^k} \right)^{pN^n}.
\end{aligned}
$$

This proves (5). In order to show that the bound tends to zero as $N^n \to \infty$ at fixed $p > 0, k \leq n$, we first note that $\frac{n!}{k!(n-k)!} \leq n^k/k!$ and thus (as $k$ is assumed fixed) it suffices to show that

$$n^k N^k \left( 1 - \frac{1}{N^k} \right)^{pN^n} \to 0 \quad \text{as} \quad N^n \to \infty.$$

Here again, we can proceed with the discussion of the two cases ($N$ bounded and $n \to \infty$, or $N \to \infty$) exactly as in the end of the proof of Prop. D.2, to conclude. □

*Remark* D.4 (Extension of Prop. D.2 and Theorems D.3 and D.1 to general $\mathcal{G}$). If, for a choice of $n \geq 2$ and $N_1, \ldots, N_n \geq 2$ we have $\mathcal{G} = \{\mathbf{G} = [G_1, \ldots, G_n] : (\forall i \leq n) 1 \leq G_i \leq N_i\}$, then $|\mathcal{G}| = N_1 \cdots N_n$ and by carefully following the same steps as in the proof of Prop. D.2 we find

$$\mathbb{P} \left\{ \begin{array}{l} \exists i \leq n, \quad \exists j \leq N_i, \\ \forall \mathbf{G} = [G_1, \ldots, G_n] \in \mathcal{G}_{train}, \\ \text{it holds } G_i \neq j \end{array} \right\} \leq \sum_{i=1}^n N_i \left( 1 - \frac{1}{N_i} \right)^{pN_1 \ldots N_n}, \tag{6}$$

Again, this tends to zero for fixed $p$ as $|\mathcal{G}| = N_1 \ldots N_n \to \infty$: to prove it, we can proceed as follows. In a first case, if $n > 2$ stays bounded, then we can discuss separately for each $1 \leq i \leq n$ whether $N_i \to \infty$ or whether it stays bounded, as done in the proof of Prop. D.2, and in either case the corresponding summand on the left of (4) tends to zero, as in case (2) from the proof of Prop. D.2. Otherwise, $n \to \infty$ and we can proceed analogously to case (1) from the end of proof of Prop. D.2.

For the extension of Thm. D.3 to general $\mathcal{G}$ we get the bound

$$
\mathbb{P} \left\{ \begin{array}{l} \exists i_1 < \cdots < i_k \leq n \\ \exists j_1, \ldots, j_k \leq N \\ \forall \mathbf{G} = [G_1, \ldots, G_n] \in \mathcal{G}_{train} \end{array} \quad (G_{i_1}, \ldots, G_{i_k}) \neq (j_1, \ldots, j_k) \right\}
$$

$$
\leq \sum_{1 \leq i_1 < \cdots < i_k \leq n} N_{i_1} \cdots N_{i_k} \left( 1 - \frac{1}{N_{i_1} \cdots N_{i_k}} \right)^{p N_1 \cdots N_n}, \tag{7}
$$

and the strategy for its proof and for the proof that this goes to zero if $k, p$ fixed and $|\mathcal{G}| = N_1 \cdots N_n \to \infty$ are a direct extension of the above.

Theorem D.1 is a direct consequence of assumption **(A)** and Theorem D.3, and thus it also directly extends to the general case.

# E    Proof that final state reconstruction allows for wider message choice freedom

We first recall the setting. Consider two fixed maps

$$
R^\star : V^C \to \mathcal{X}, \quad S^\star : \mathcal{D} \subseteq \mathcal{X} \to V^C,
$$

and let $\mathcal{R}_{[i]} : V^C \to V^i$ be the restriction to the first $i$ tokens, i.e. in previous notation, for $m = (m_1, \ldots, m_C) \in V^C$ we set $\mathcal{R}_{[i]}(m) := m_{[i]} = (m_1, \ldots, m_i)$.

For $R^\star, S^\star$ as above and $S : \mathcal{D} \subseteq \mathcal{X} \to V^C$ we set, for $x \in \mathcal{D}$

$$
d_i^S(x) := d_{\mathcal{X}}(R^\star(S^\star(x)_{[i]}), R^\star(S(x)_{[i]})) = d_{\mathcal{X}}(R^\star(\mathcal{R}_{[i]}(S^\star(x))), R^\star(\mathcal{R}_{[i]}(S(x)))).
$$

We also defined the following spaces:

$$
\mathcal{S}_{full}^\epsilon := \left\{ S : \mathbb{E}_x \left[ \frac{1}{C} \sum_{i=1}^C d_i^S(x) \right] < \epsilon \right\}, \quad \mathcal{S}_{final}^\epsilon := \left\{ S : \mathbb{E}_x \left[ d_C^S(x) \right] < \epsilon \right\}. \tag{8}
$$

Our main result here is the following:

**Proposition E.1.** *With the above notations, we have the following*

*1. For $i < C$ and any choice of $M, \epsilon > 0$, the restrictions on $i$-token sub-messages of $S(x)$ by requiring $S \in \mathcal{S}_{full}^M$ are more restrictive than those imposed by requiring $S \in \mathcal{S}_{final}^\epsilon$, i.e. we have $\mathcal{R}_{[i]}(\mathcal{S}_{full}^M) \subset \mathcal{R}_{[i]}(\mathcal{S}_{final}^\epsilon)$.*

*2. For the full message case $i = C$ in general we have $\mathcal{S}_{full}^\epsilon \subseteq \mathcal{S}_{final}^{C\epsilon}$. If we further assume that $\mathbb{E}_x[d_j^S(x)]$ is non-increasing in $j$ then we have the stronger inclusion $\mathcal{S}_{full}^\epsilon \subseteq \mathcal{S}_{final}^\epsilon$.*

*Proof.* Item 1 is direct, as no restriction on intermediate sub-messages $S(x)_{[i]} = \mathcal{R}_{[i]}(S(x))$ by the requirement $S \in \mathcal{S}_{final}^\epsilon$.

For item 2, we observe that

$$
\frac{1}{C} d_C^S(x) \leq \frac{1}{C} \sum_{i=1}^C d_i^S(x),
$$

and by taking expectation we find that $\mathbb{E}_x \left[ \frac{1}{C} \sum_i d_i^S(x) \right] < \epsilon$ implies $\mathbb{E}_x[d_C^S(x)] \leq C\epsilon$, and thus $\mathcal{S}_{full}^\epsilon \subseteq \mathcal{S}_{final}^{C\epsilon}$, which gives the first part of the statement.

For the second part of item 2 we note that if $\mathbb{E}_x[d_i^S(x)] \geq \mathbb{E}_x[d_C^S(x)]$ for all $i \leq C$ we get

$$\mathbb{E}_x\left[d_C^S(x)\right] \leq \min_{1 \leq j \leq C} \mathbb{E}_x[d_j^C(x)] \leq \frac{1}{C}\sum_{i=1}^{C}\mathbb{E}_x\left[d_i^S(x)\right] = \mathbb{E}_x\left[\frac{1}{C}\sum_{i=1}^{C}d_i^S(x)\right],$$

and thus $\mathcal{S}_{full}^\epsilon \subseteq \mathcal{S}_{final}^\epsilon$ as desired. □

The freedom of exploration ensured by item 1 of the proposition is our main motivation: by only constraining the reconstruction of the full message, we allow much more freedom for the encoding strategy of $S(x)$ at earlier sub-messages at each iteration.

Item 2 of Prop. E.1 shows that the two losses in the definition of $\mathcal{S}_{full}, \mathcal{S}_{final}$ give comparable guarantees. At the beginning of iterations we will need to use the general bound with $\epsilon/C$, and as $R^\star$ will have been trained on more and more messages during the iteration, the hypothesis of $\mathbb{E}_x[d_i^S(x)]$ non-increasing will become true allowing for the sharper bound with the same value of $\epsilon$, as $R^\star$ (an LSTM taking as input successive message tokens) will become efficient in using sub-messages of pairs $(m, m') = (S(x), S^\star(x))$ to gradually distill information allowing to distinguish them for a large set of $S, x$.

# F    Regularization to counteract Holistic Encodings

In this section, we include rigorous results useful to justify the regularization of our **Imitation** phase, aimed at balancing **compressibility** and **efficiency**.

This corresponds to the balance between compositional and parsimonious communication, a general theme in classical IL literature (see e.g. [35, 37]). It seems useful to specify more precise definitions in our generative setting, in order to help justify our architecture choices.

Our aim in this section is threefold:

1. In §F.1 we set rigorous definitions of Holistic Encodings within the setting of §2.1 and §2.2.
2. In §F.2 we check that **compressibility** is required for lowest Kolmogorov Complexity encodings.
3. We verify that optimizing for just **expressivity** and **efficiency** on our training set, will produce Holistic Encodings, and thus is not satisfactory for reconstructing $\mathcal{G}$ (see §F.3);
4. We give a few possible options on how to enrich the loss functions in our IL setup, aimed at establishing a better balance between **efficiency** and **compressibility** (see §F.4).

### F.1    Definitions of Holistic and Compositional encodings, and role of the compressibility condition

In the classical IL setup [35, 37], a holistic system of communication is defined as an encoding $\mathcal{G} \to \mathcal{M}$ that maps $\mathcal{G} \ni \mathbf{G} \mapsto \mathbf{G}' \in \mathcal{M}$ injectively but disregards the compositional structure of the "language" to be encoded, i.e. in our case, without respecting the structure of $\mathbf{G}$. In our generation task formalization, a difference is that the relevant encoding map now factors through the generation map as $\mathcal{G} \xrightarrow{\text{GenX}} \mathcal{X} \xrightarrow{S} \mathcal{M}$. Now the generation map GenX is considered fixed, thus the definition refers to only $S$ and is the following:

**Definition F.1.** In the setting from §2.1 and §2.2, sender $S : \mathcal{X} \to \mathcal{M}$ determines a *holistic encoding* if it satisfies **expressivity** but the messages associated to data generated as GenX($\mathbf{G}$) do not have a compositional structure in terms of generating factors $\mathbf{G}$, as specified by Definition F.2 below.

Note that, as in [35, 37], Definition F.1 of holistic encodings refers to "respecting compositional structure", which requires another definition. The definition of compositional encodings *crucially depends on how complex we need/want to assume* $\mathbf{G}$ *to be*. In this paper, for the sake of concreteness, we restricted to the simple case of $\mathbf{G}$ being formed as an element of a product space of finite cardinality, i.e., an $n$-ple of independent factors $\mathbf{G} = [G_1, \ldots, G_n]$ in which each $G_i$ can take finitely many values. We thus use a straightforward toy definition of compositional encodings, requiring that each of the $G_i$ is separately encoded as a string and these strings are concatenated together to give the message that encodes $\mathbf{G}$:

**Definition F.2.** In the setting of §2.1 and §2.2, assume that generating factors $\mathbf{G} = [G_1, \ldots, G_n]$ have $G_i \in [1 : N_i]$ for some $N_1, \ldots, N_n \geq 2$. Then sender $S : \mathcal{X} \to \mathcal{M} = V^C$ will be defined to *respect the structure of generating factors* $\mathbf{G}$ if for each $1 \leq i \leq n$ there exist an injective mapping $E_i : [1 : N_i] \to V^{C_i}$, for which

$$\forall \mathbf{G} \in \mathcal{G}, \ S(\mathsf{GenX}(\mathbf{G})) = E_1(G_1) \cdots E_n(G_n) \quad \text{and} \quad \sum_{i=1}^{n} C_i = C.$$

The **compressibility** condition on the sender-receiver protocol just requires that the protocol be not holistic, and thus it is fully specified via definitions F.1 and F.2. Furthermore, Definition F.2 fully specifies in which way the map $\mathcal{G} \to \mathcal{M}$ successfully approximates an isomorphic reconstruction of $\mathcal{G}$, if it satisfies conditions **compressibility** and **expressivity**.

## F.2 Encodings optimizing for only expressivity and efficiency have Kolmogorov Complexity much higher than compositional ones

By the classical learning theory principle, if our sender generates encodings with lower complexity, these should be easier to learn by the receiver, in the sense that the receiver will tend to generalize with more ease form good accuracy on the training set to similar accuracy on the test set.

As a manageable measure of complexity we will use Kolmogorov Complexity (KC), and we verify that in our setting, encodings satisfying **expressivity** and **efficiency** will necessarily have higher KC than compositional ones. In particular, the receiver will have more trouble reconstructing from such encodings, justifying our push to enforce the **compressibility** condition.

Encodings of $\mathcal{D}_{train}$ satisfying **expressivity** and **efficiency** include all injective maps

$$\mathsf{enc} : \mathcal{D}_{train} \to V^{C_\star}, \quad C_\star = \lceil \log_{|V|} |\mathcal{D}_{train}| \rceil, \tag{9}$$

in which $\lceil a \rceil$ denotes the smallest integer larger than real number $a$. For a random map enc as above, the encoding to be Kolmogorov-irreducible, so that the expected KC of a random enc would be up to constant factor the one corresponding to explicitly enumerating the $|\mathcal{D}_{train}|$ strings of length $C_\star$ in alphabet $V$ that describe the encodings of each element of the training set. Each such string thus requires $\approx C_\star \log_2 |V|$ bits. Thus if $U$ is the uniform distribution over injective maps (9) then we have

$$\mathbb{E}_{\mathsf{enc} \sim U}[KC(\mathsf{enc})] \approx |\mathcal{D}_{train}| C_\star \log_2 |V| \approx |\mathcal{D}_{train}| \log_2 |\mathcal{D}_{train}|. \tag{10}$$

On the other hand, to specify a compositional encoding comp as in Definition F.2 requires only assigning a string of length $C_i$ in alphabet $V$ for every generating factor $G_i$, where we can take $C_i = \lceil \log_{|V|} N_i \rceil$. Thus such string requires $\approx \log_2 N_i$ bits, and the whole encoding has KC given by

$$KC(\mathsf{comp}) \approx \sum_{i=1}^{n} \log_2 N_i = \log_2 |\mathcal{D}_{train}|. \tag{11}$$

If we assume that $|\mathcal{D}_{train}|/|\mathcal{D}| = p \in (0, 1)$ and we take $|\mathcal{D}| \to \infty$, then we get

$$\frac{\text{r.h.s. of (10)}}{\text{r.h.s. of (11)}} = \frac{p|\mathcal{D}|(\log_2 p + \log_2 |\mathcal{D}|)}{\log_2 |\mathcal{D}|} \sim p|\mathcal{D}| \quad \text{as } |\mathcal{D}| \to \infty. \tag{12}$$

The computation (12) shows that

**Proposition F.3.** *Under the same hypotheses on $\mathcal{G}$ as in the previous section, consider the limit $|\mathcal{D}| \to \infty$ and assume that $p = |\mathcal{D}_{train}|/|\mathcal{D}|$ stays bounded away from zero in the limit. Then the expected Kolmogorov complexity of encodings satisfying **expressivity** and **efficiency** becomes overwhelmingly larger than the one of compositional encodings:*

$$\frac{\mathbb{E}_{\mathsf{enc} \sim U}[KC(\mathsf{enc})]}{KC(\mathsf{comp})} \sim p|\mathcal{D}| \to \infty.$$

## F.3 Encodings that optimize efficiency under expressivity over $\mathcal{D}_{train}$ are holistic

First, we clarify the definitions:

- The requirement of "**expressivity** over $\mathcal{D}_{train}$" means that the encodings are injective over $\mathcal{D}_{train}$.
- The requirement of "**efficiency** over $\mathcal{D}_{train}$" means that the encoding maps $\mathcal{D}_{train}$ into $V^{C_\star}$ for the minimum value of $C_\star$.

Similarly to previous subsection (see (9)), keeping in mind that $|\mathcal{D}_{train}| = p|\mathcal{D}| = p|\mathcal{G}|$, with fixed $p \in (0, 1)$, we have

$$C_\star = \lceil \log_{|V|} |\mathcal{D}_{train}| \rceil = \lceil \log_{|V|} p|\mathcal{G}| \rceil \leq \lceil \log_{|V|} |\mathcal{G}| \rceil - \lfloor |\log_{|V|} p| \rfloor. \tag{13}$$

For compositional encodings of $\mathbf{G} = [G_1, \ldots, G_n]$ the encoding will require to set apart separate message sub-strings of length $C_i = \lceil \log_{|V|} N_i \rceil$ for factor $G_i$ (see Def. F.2), thus the required length for the encoding will be

$$C_0 := \sum_{i=1}^{n} \lceil \log_{|V|} N_i \rceil \overset{(\star)}{\geq} \left\lceil \sum_{i=1}^{n} \log_{|V|} N_i \right\rceil = \lceil \log_{|V|} |\mathcal{G}| \rceil \overset{(\star\star)}{\geq} C_\star + \lfloor |\log_{|V|} p| \rfloor, \tag{14}$$

in which $(\star\star)$ follows from (13).

We now discuss when the inequality signs in (14) are sharp or not, since sharp inequality $C_0 > C_\star$ implies our claim that optimizing for **efficiency** under **expressivity** implies non-compositional (i.e., holistic) encodings:

1. If we use relatively small $|V| \lesssim 10$ (assuming commonly used values $p \lesssim 0.1$), we will have

$$|V| \leq \frac{1}{p}, \tag{15}$$

and thus $\log_{|V|} p \leq -1$, and thus the second inequality $(\star\star)$ in (14) is guaranteed to be sharp, showing the desired strict inequality $C_0 > C_\star$ independently of the sizes of factor ranges $N_i, i = 1, \ldots, n$.

2. If the gaps between $\log_{|V|} N_i$ and the lowest integer larger or equal to it, sum to a value of at least 1, i.e.,

$$\sum_{i=1}^{n} \left( \lceil \log_{|V|} N_i \rceil - \log_{|V|} N_i \right) \geq 1, \tag{16}$$

then $(\star)$ becomes sharp, again guaranteeing $C_0 > C_\star$ as desired. This could be probable to happen in our setting, and becomes more likely for large $|V| > 1/p$ for which (15) fails, **given that the $N_i$ are supposed to be unknown**, so it is likely that for large $|V| > 1/p$ for a few of the $N_i$ the gaps in (16) are nontrivial. For example, (16) holds true for $|V| > 1/p \geq 10$ if, say, at least two of the $G_i$ are binary factors, so that $N_i = 2$.

In summary we have the following:

**Proposition F.4.** *Under the same running assumptions over $\mathcal{G}$, $p$, we have that optimizing only for efficiency of encodings **efficiency** under injectivity constraints **expressivity** over $\mathcal{D}_{train}$ is guaranteed to produce holistic (i.e., non-compositional) encodings if one or both of the conditions (15) and (16) holds.*

### F.4 Possible choices of encoded message regularization

As seen above, if we just optimize for **efficiency** and **expressivity** then we are likely to get holistic encodings (Prop. F.4), which will then make it hard for the receiver to generalize the decoding strategy outside the training set due to having overwhelmingly higher expected complexity (Prop. F.3).

This proves that it is important to incentivize condition **compressibility**, i.e., to push encoding strategies away from overly efficient holistic encodings. In this section we discuss a few approaches to do this in practice, explaining our choice of **compressibility** regularization.

1. **Entropy maximization.**

**Background and standard approach.** Introduced by [69] for use in policy-based reinforcement learning, entropy maximization is known to enforce exploration in deterministic policies and avoid early convergence to single output choices for learned policies. Specifically, in a learning setup with input $x$, output $y$ and policy parameters $W$, the authors define the following estimator for fixed output value $\xi$:

$$h(\xi, W, x) = -\ln Pr(y = \xi | W, x), \tag{17}$$

such that if we take the expectation over $\xi$ we get:

$$\mathbb{E}[h(\xi, W, x) | W, x] = -\sum_{\xi} Pr(y = \xi | W, x) \ln Pr(y = \xi | W, x), \tag{18}$$

which is the entropy of the network. Therefore $h$ is an unbiased estimator of the entropy. They also note that, if $w_i$ and $x_i$ with $i \in U$ represent the weights and pattern of single neurons in a feed forward network, $\xi_i$ the $i_{th}$ position in a $n_u$-tuple $\xi$, and let $g_i$ be the probability density describing $y_i$, i.e:

$$g_i(\xi, w^i, x^i) = Pr(y_i = \xi | w^i, x^i), \tag{19}$$

then, for any feed forward network,

$$Pr(y = \xi | W, x) = \prod_{i \in U} g_i(\xi_i, w^i, x^i), \tag{20}$$

and therefore

$$-\ln Pr(y = \xi | W, x) = -\sum_{i \in U} g_i(\xi_i, w^i, x^i) = h(y, W, X). \tag{21}$$

In fact, this method can be extended onto sequence prediction models, and even IL, such as in [55].

**Naive adaptation to our setting is not practical.** We take $x \in \mathcal{X}$ and $\xi \in \mathcal{M} = V^C$ and recall that in our notation $m_t$ indicates the $t$-th token of message $m$, and $m_{[t]}$ is the notation for the initial segment $m_1, m_2, \ldots, m_t$, so that in particular $m_{[C]} = m$.

Importantly, in our setting, to avoid high variance in estimation due to the large message space (see §G), we approximate the probability distribution of the RNN-produced message token at position $t \leq C$, denoted here $p(m_t | x, m_{[t-1]})$, as a $\delta$ distribution over the vocabulary $V$ by using the Gumbel-softmax trick:

$$(\exists v \in V), \quad Pr(m_t \mid x, W, m_{[t-1]}) \approx \begin{cases} 1, & \text{if } m_t = v \\ 0, & \text{if } m_t \in V \setminus \{v\}, \end{cases} \tag{22}$$

and therefore:

$$Pr(m = \xi | W, x) = \prod_{t=1}^{C} Pr(m_t = \xi_t | W, x, m_{[t-1]}). \tag{23}$$

Note that due to (22), for all but a single one of the $|V|^C$ possible values of $\xi$ the quantity (23) is $\approx 0$, and thus $h$ becomes a poor estimator for the entropy of $p(m_t | x, m_{[t-1]})$, as we get

$$h(m, W, x) = -\ln \sum_{t=1}^{C} Pr(m_t = \xi_t | x, W, m_{[t-1]}) \approx \infty. \tag{24}$$

In particular, it becomes clear that *optimizing this $h$ becomes unfeasible with gradient methods.*

**Our proposal for regularization.** Our model can be seen as a different approximation of the entropy of the message. Instead of using the policy weights as probability values, we use the law of large numbers to estimate the probability of each message. For a batch $\mathcal{B} \subset \mathcal{X}$ consisting of $N_{batch}$ training examples and for given sender encoding protocol $S_\phi$:

$$\text{For a sufficiently large } N_{batch}: \quad \Pr(m = \xi \mid x, \phi) \approx \frac{|\{x \in \mathcal{B} \mid S_\phi(x) = \xi\}|}{N_{batch}} \tag{25}$$

Yet again, due to the exponential size of the message space, utilizing this probability to estimate the entropy is still not practical. **We will assume approximate independence between symbol positions**, which allows to get the stronger approximation:

$$\Pr(m_t = v \mid x, \phi) \approx \frac{|\{x \in \mathcal{B} \mid S_\phi(x)_t = v\}|}{N_{batch}}, \tag{26}$$

as well as the additivity of entropy to be applied to (18):

$$
\begin{aligned}
H(Pr(m|x,\phi)) \quad &:= \quad -\sum_{\xi \in V^C} Pr(m = \xi \mid x, \phi) \ln Pr(m = \xi \mid x, \phi) \\
&= \quad -\sum_{t=1}^{C} \sum_{v \in V} Pr(m_t = v \mid x, \phi) \ln Pr(m_t = v \mid x, \phi). \tag{27}
\end{aligned}
$$

Then we can perform the following computations, in which we let $m, m'$ be two i.i.d. copies of $m$:

$$
\begin{aligned}
H(Pr(m \mid x, \phi)) &\overset{(27)}{=} -\sum_{t=1}^{C} \sum_{v \in V} Pr(m_t = v \mid x, \phi) \ln Pr(m_t = v \mid x, \phi) \\
&\overset{\substack{(-x \ln x \geq x - x^2 \\ \text{for } x \leq 1)}}{\geq} \sum_{t=1}^{C} \sum_{v \in V} \left[ Pr(m_t = v \mid x, \phi) - Pr(m_t = v \mid x, \phi)^2 \right] \\
&\overset{(\star)}{=} \sum_{t=1}^{C} \left( \sum_{v,v' \in V} Pr(m_t = v \mid x, \phi) Pr(m'_t = v' \mid x, \phi) \right. \\
&\qquad\qquad \left. - \sum_{v \in V} Pr(m_t = v \mid x, \phi)^2 \right) \\
&= \sum_{t=1}^{C} \sum_{v \neq v' \in V} Pr(m_t = v \mid x, \phi) Pr(m'_t = v' \mid x, \phi) \\
&= \sum_{t=1}^{C} Pr(m_t \neq m'_t) = \mathbb{E}[d_H(m, m')] \\
&\overset{(26)}{\approx} \mathbb{E}_{x,x' \in \mathcal{B}}[d_H(S_\phi(x), S_\phi(x'))], \tag{28}
\end{aligned}
$$

In the above step $(\star)$ we used that

$$
\begin{aligned}
\sum_{v \in V} Pr(m_t = v \mid x, \phi) \quad &= \quad 1 = \left( \sum_{v \in V} Pr(m_t = v \mid x, \phi) \right)^2 \\
&= \quad \sum_{v,v' \in V} Pr(m_t = v \mid x, \phi) Pr(m'_t = v' \mid x, \phi).
\end{aligned}
$$

The consequence of bound (28) is that maximizing the pairwise Hamming distance between predicted messages in a batch also maximizes a lower bound to the approximate entropy of the message space, justifying the choice of optimizing this quantity (which is easier to i include in practice) for favoring **compressibility**.

2. **Contrastive Learning losses.** In our framework, PDM can be seen as a form of contrastive loss, which we now show how to connect to standard contrastive losses from previous works.

Contrastive loss functions such as the triplet loss [63] and NT-Xent [11] work by defining positive and negative image pairs (usually through image augmentation), and aiming to minimize the distance between embeddings of positive pairs and maximize the distance between negative pairs. Specifically, instead of purposefully sampling negative samples for every $x$, NT-Xent produces a single pair of samples for each $x$ in a mini-batch $N$ by

augmenting each image. Each image in the batch therefore has $2N-1$ negative pairs. They use the cosine similarity $sim(u,v) = u^T v/\|u\|\|v\|$ and some temperature value $\tau$ to define the following loss between positive pair embeddings $z_i$ and $z_j$:

$$\ell_{i,j} = -\log \frac{\exp(sim(z_i, z_j)/\tau)}{\sum_k^{2N} \exp(sim(z_i, z_k)/\tau)} \tag{29}$$

For simplicity, let $\tau = 1$ and thus

$$\ell_{i,j} = -sim(z_i, z_j) + \log \sum_k^{2N} \exp(sim(z_i, z_k)). \tag{30}$$

Here, we apply the same ideas but take this loss over an unaugmented batch, i.e the only application of this loss is for $z_i$ against itself. In this case it reduces to:

$$\ell_i = -1 + \log \sum_k^N \exp(sim(z_i, z_k)) \tag{31}$$

In our representations, the embedding vectors from a batch $z_i, i = 1, \ldots, N$ are $V \times C$ matrices whose columns are approximately one-hot vectors that represent the tokens of a message. For one-hot matrices $z_i, z_k$ of this form, the dot product $z_i^T z_k$ is equivalent to counting the amount of matching rows of $z_i, z_k$, i.e, the un-normalized Hamming distance of the associated message strings $m_i, m_k \in V^C$. Since strings have length $C$ and $\|z_i\|^2 = z_i^T z_i \simeq C$, we thus get:

$$
\begin{aligned}
sim(z_i, z_k) &= tr(z_i^T z_k)/\|z_i\|\|z_k\| \simeq \frac{1}{C} \sum_{l=1}^C \mathbb{1}[\arg\max_j((z_i)_{l,j}) = \arg\max_j((z_k)_{l,j})] \\
&\simeq 1 - d_H(m_i, m_k),
\end{aligned}
$$

in which $d_H$ is the (normalized by $1/C$) Hamming distance. We can replace this expression in the unaugmented NT-Xent loss:

$$
\begin{aligned}
\ell_i &= -1 + \log \sum_{k=1}^N \exp(1 - d_H(m_i, m_k)) \\
&= -1 + N - \log \sum_{k=1}^N \exp(d_H(m_i, m_k)) \\
&\overset{\substack{\text{(Jensen} \\ \text{inequality)}}}{\leq} -1 + N - \log N - \frac{1}{N} \sum_{k=1}^N d_H(m_i, m_k).
\end{aligned}
$$

Then, taking the mean loss over the full batch we get:

$$\frac{1}{N} \sum_{i=1}^N \ell_i \leq -1 + N - \log N - \frac{1}{N^2} \sum_{i=1}^N \sum_{k=1}^N d_H(m_i, m_k). \tag{32}$$

We then include in the loss only the last term, i.e. the average of $d_H(m_i, m_k)$, which is the only term that actually depends on the batch elements, which forms an approximate upper bound on the NT-Xent loss.

Note that our approximation of $sim(z_i, z_k)$ by $1 - d_H(m_i, m_k)$, on which the above is based, becomes sharp when the $z_i$ have actual one-hot entries, thus the approximation becomes better as our model has high accuracy. Furthermore, Jensen's inequality, responsbile for the inequality between the term $\ell_i$ from NT-Xent and our metric, becomes closer to equality messages $m_1, \ldots, m_N$ become uniformly spread across $V^C$, which is the case for minimizers of our metric as well. This means that, at least heuristically speaking, we can consider our metric as having similar minimizers to the NT-Xent regularization.

A precise matching lower bound to (32) is left for future work.

## G  Justification of our imitation step starting from classical probabilistic reconstruction games

We start by recalling classical probabilistic reconstruction games, after which we adapt the formulation to the setting of generative tasks, and we show that with a few justified simplifications we arrive to our formulation from the main paper.

### G.1  Classical probabilistic reconstruction games

Recall that in reconstruction games, the two agent types are a teacher and student (or speaker and listener, or sender and receiver), parameterized by respectively $\theta, \phi$. The game consists in the teacher observing a signal $x$ distributed according to a probability distribution $p$, from which it produces a randomized message $m$ whose distribution conditional on $x$ is obtained according to policy $\pi_\theta(\cdot|x)$. The student or listener, observes message $m$ and produces a reconstructed signal distributed according to a conditional probability $\rho_\phi(\cdot|m)$. The game dynamics can be modeled by stipulating that together, teacher and student minimize the log-likelihood loss function

$$\mathcal{L}_{\theta,\phi} = -\mathbb{E}_{x \sim p, m \sim \pi_\theta(\cdot|x)} \left[ \log \rho_\phi(x|m) \right] \tag{33}$$

in which $x$ represents the signal and $m$ the message used to reconstruct it, the speaker produces a reinforcement learning policy $\pi_\theta(\cdot|x)$ and the listener produces a reconstruction function $\rho_\phi(x|m)$.

### G.2  Reconstruction games in generative tasks

While the above formulation (33) is formalized using the log-likelihood between distributions, for our generative task, the reconstruction modeled slightly differently: the task is to optimize, with high probability, the reconstruction error, measured by a distance between the signal $x$ and its reconstruction $x'$, denoted $d(x, x')$. Here the notation for $x, x'$ and $d(x, x')$ can be (ab)used to represent two alternative settings: either (a) $x, x'$ are vectors in the autoencoder latent space and $d(x, x')$ is a discrepancy (e.g. $d(x, x') = |x - x'|^p$) between them or (b) $x, x'$ are reconstructed images and $d(x, x')$ is a discrepancy (e.g. based on some ad-hoc norm or otherwise) between them. Thus we have the probabilistic error function

$$\mathcal{L}^{\mathbb{P}}_{int}(\omega, \varphi) := \mathbb{E}_{x \sim p, m \sim \pi_\varphi(\cdot|x), x' \sim \rho_\omega(\cdot|m)} d(x, x'). \tag{34}$$

The loss (34) is akin to a $d$-based Wasserstein distance analogue of (33).

### G.3  The problem reduces to the case of a deterministic student

Note that when $d(x, x) = |x - x'|^2$ for some Euclidean norm (or more generally, when $d$ is a convex function of such norm), then the minimization of (34) over distributions $\rho_\omega$ at fixed $m, \varphi$ must be achieved by

$$x^\varphi_m \in \mathrm{argmin}_{x'} \mathbb{E}_{x \sim p_\varphi(\cdot|m)} d(x, x'), \tag{35}$$

where if the set of $x$ is finite then $p_\varphi$ is the probability defined by

$$p_\varphi(x'|m) = \frac{p(x')\pi_\varphi(m|x')}{\sum_y p(y)\pi_\varphi(m|y)}, \tag{36}$$

and an analogous expression with integrals replacing the sum holds in general.

Under the above convexity hypotheses, the choice of $x^\varphi_m$ in (35) is unique, and for $d(x, x') = |x - x'|^2$ it coincides with the barycenter of $p_\varphi$. In this case, $\rho_\omega(\cdot|m)$ can be taken to be the Dirac distribution concentrated at $x^\varphi_m$, and optimization of (34) can be achieved by a deterministic reconstruction function $R_\omega(m)$, whose optimum value for given $\varphi$ will be $x^\varphi_m$ from (35). When reducing to optimizing only amongst deterministic $R_\omega$, the loss (34) rewrites in the simplified form

$$\mathcal{L}_{int}(\omega, \varphi) = \mathbb{E}_{x \sim p, m \sim \pi_\varphi(\cdot|x)} d(x, R_\omega(m)). \tag{37}$$

### G.4 Straight-Trough Gumbel-Softmax (STGS) trick

As in [25] and subsequent works, the optimization of policy $\pi_\varphi$ (minimization over $\varphi$) can in theory be done via policy gradient optimization methods such as REINFORCE [70], however, due to the combinatorial explosion of the number of messages this can be very compute-intensive if calculated directly, or produces high variance estimators. We thus follow the straight-trough Gumbel-softmax trick [26], in which the probabilistic interpretation of $\pi_\varphi$ is implicit in the network architecture, and the architecture itself is deterministic and directly differentiable. In this case, we can thus pass to the deterministic description as $m = S_\phi(x)$, replacing the probabilistic one $m \sim \pi_\varphi(\cdot|m)$, and we obtain the updated form of (37) as

$$\mathcal{L}_{int}(\omega, \phi) = \mathbb{E}_{x \sim p} d(x, R_\omega(S_\phi(x))), \tag{38}$$

which for $d(x, x') = |x - x'|$ is very similar to our actual interaction loss (1):

$$\mathcal{L}_{int}(\omega, \phi) = \mathbb{E}_{x \in \mathcal{X}} \left[ |x - R_\omega(S_\phi(x))| \right]. \tag{39}$$

# H    Example Generation: Reconstructions

The following reconstruction, Figure 5, examples were generated with the CELEBI model using PDM regularization with $\beta = 1$ and $\lambda = 1.5$. In each row, the leftmost image corresponds to the original input, while the subsequent images represent the predicted reconstructions $x_i = R_\omega(S_\phi(x)_{[1:i]})$ obtained from progressively longer message prefixes with the VAE backbone.

We observe that the reconstructions do not merely improve in pixel-level fidelity but rather exhibit semantic refinements across successive steps — variations in floor hue, object color, shape, or viewing angle. This progression suggests that the message space encodes the underlying generative factors $\mathcal{G}$ and supports a degree of compositional structure, where messages carry disentangled information.

**Qualitative comparison across methods.**    To further assess the representational properties of CELEBI, we qualitatively compare reconstructions across the different methods proposed in this work (see Figures 6 and 7). As illustrated in these figures, CELEBI reaches semantically faithful reconstructions significantly earlier in the decoding sequence than the baseline methods.

Moreover, the differences between reconstructions are more semantically pronounced than in baseline models, which often show subtler variations mainly related to texture or brightness. This qualitative evidence supports our hypothesis that CELEBI facilitates a more structured message space, enabling representations that are not only more compact but also more semantically coherent.

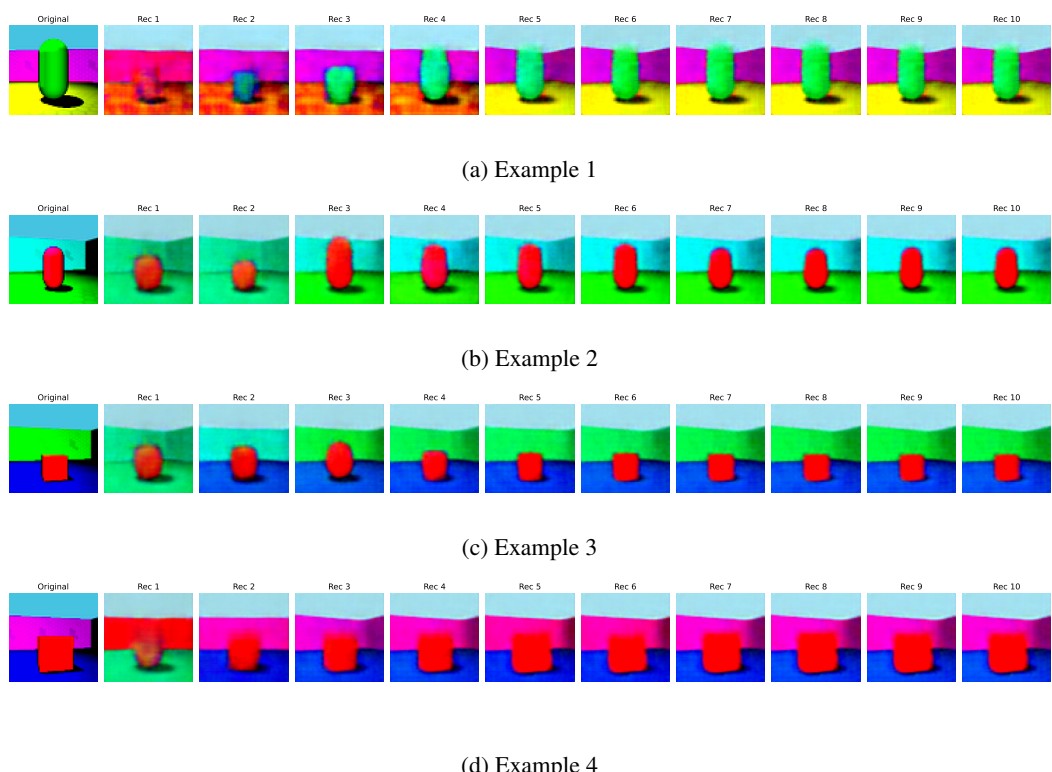

(a) Example 1

(b) Example 2

(c) Example 3

(d) Example 4

Figure 5: Progressive image reconstructions obtained at each decoding step by the receiver as it processes successive symbols of the message.

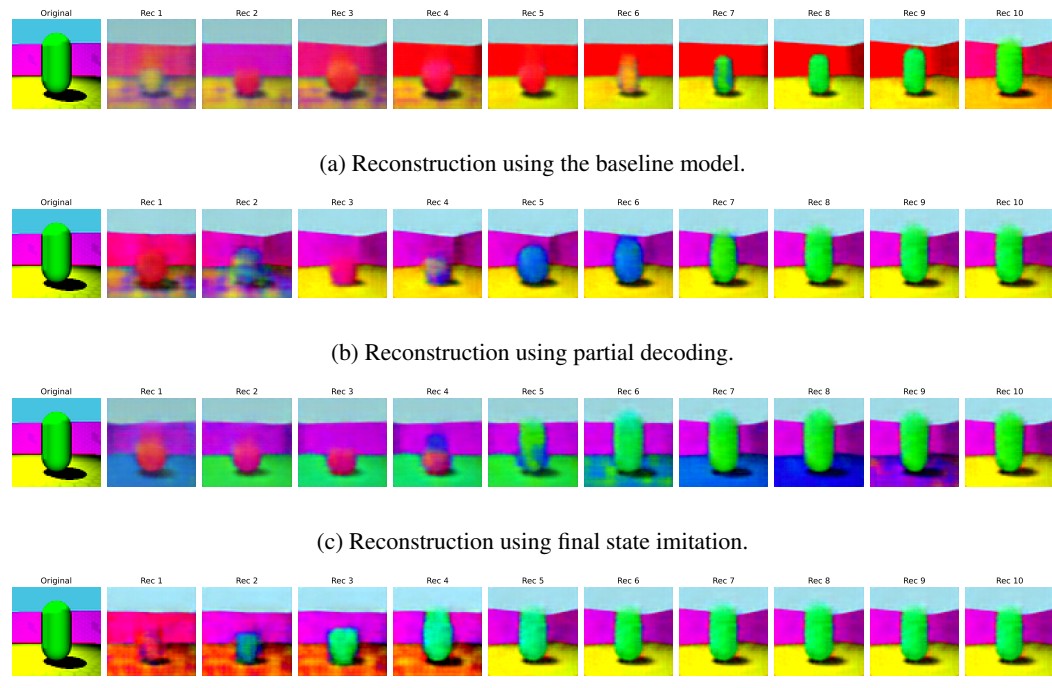

(a) Reconstruction using the baseline model.

(b) Reconstruction using partial decoding.

(c) Reconstruction using final state imitation.

(d) Reconstruction using PD+FiSI+PDM.

Figure 6: Reconstruction for the same example with different methods.

# I    Additional Language Metrics

Table 3: Comparison of proposed methods using language variation metrics from [13].

|  |  | Synonymy ↓ | Homonymy ↓ | Freedom ↓ | Entanglement ↓ |
|---|---|---|---|---|---|
| **Interaction** | Full message reconstruction, no IL (Baseline) | $0.563 \pm 0.026$ | $0.695 \pm 0.015$ | $0.569 \pm 0.026$ | $0.831 \pm 0.006$ |
|  | Progressive Decoding (PD, ours) | $0.640 \pm 0.022$ | $0.737 \pm 0.013$ | $0.646 \pm 0.013$ | $0.786 \pm 0.007$ |
| **Imitation** | Message imitation (baseline) | $0.601 \pm 0.039$ | $0.714 \pm 0.027$ | $0.613 \pm 0.039$ | $0.822 \pm 0.010$ |
|  | Full state imitation | $0.597 \pm 0.017$ | $0.734 \pm 0.021$ | $0.604 \pm 0.018$ | $0.824 \pm 0.007$ |
|  | Final-State Imitation (FiSI, ours) | $0.633 \pm 0.038$ | $0.751 \pm 0.018$ | $0.640 \pm 0.038$ | $0.842 \pm 0.007$ |
| **Regularization** | PD+FiSI+KoLeo $\lambda = 1.5$ | $0.594 \pm 0.027$ | $0.739 \pm 0.013$ | $0.598 \pm 0.027$ | $0.805 \pm 0.007$ |
|  | PD+FiSI+PDM (ours) $\lambda = 1.5$ | $0.654 \pm 0.013$ | $0.753 \pm 0.008$ | $0.658 \pm 0.013$ | $0.780 \pm 0.008$ |

(a) Shapes3D

|  |  | Synonymy ↓ | Homonymy ↓ | Freedom ↓ | Entanglement ↓ |
|---|---|---|---|---|---|
| **Interaction** | Full message reconstruction, no IL (Baseline) | $0.595 \pm 0.053$ | $0.597 \pm 0.061$ | $0.597 \pm 0.052$ | $0.852 \pm 0.021$ |
|  | Progressive Decoding (PD, ours) | $0.484 \pm 0.044$ | $0.486 \pm 0.056$ | $0.486 \pm 0.044$ | $0.786 \pm 0.025$ |
| **Imitation** | Message imitation (baseline) | $0.367 \pm 0.034$ | $0.393 \pm 0.042$ | $0.369 \pm 0.034$ | $0.837 \pm 0.015$ |
|  | Full state imitation | $0.362 \pm 0.045$ | $0.424 \pm 0.052$ | $0.364 \pm 0.044$ | $0.832 \pm 0.018$ |
|  | Final-State Imitation (FiSI, ours) | $0.324 \pm 0.020$ | $0.393 \pm 0.018$ | $0.328 \pm 0.020$ | $0.830 \pm 0.008$ |
| **Regularization** | PD+FiSI+KoLeo $\lambda = 1.5$ | $0.365 \pm 0.011$ | $0.338 \pm 0.019$ | $0.367 \pm 0.011$ | $0.763 \pm 0.005$ |
|  | PD+FiSI+PDM (ours) $\lambda = 1.5$ | $0.347 \pm 0.012$ | $0.333 \pm 0.010$ | $0.349 \pm 0.013$ | $0.756 \pm 0.005$ |

(b) MPI3D

In addition to the metrics presented in the main text of this work, we attempted to measure other metrics for qualitative linguistic variation. We were not able to find many such metrics in the literature, and thus restrict to the ones presented in [13]. Importantly, since this reference had a slightly different focus, variation in this context will not refer to the evolution of the language in time, but rather a departure from regularity, which masks an underlying compositional structure.

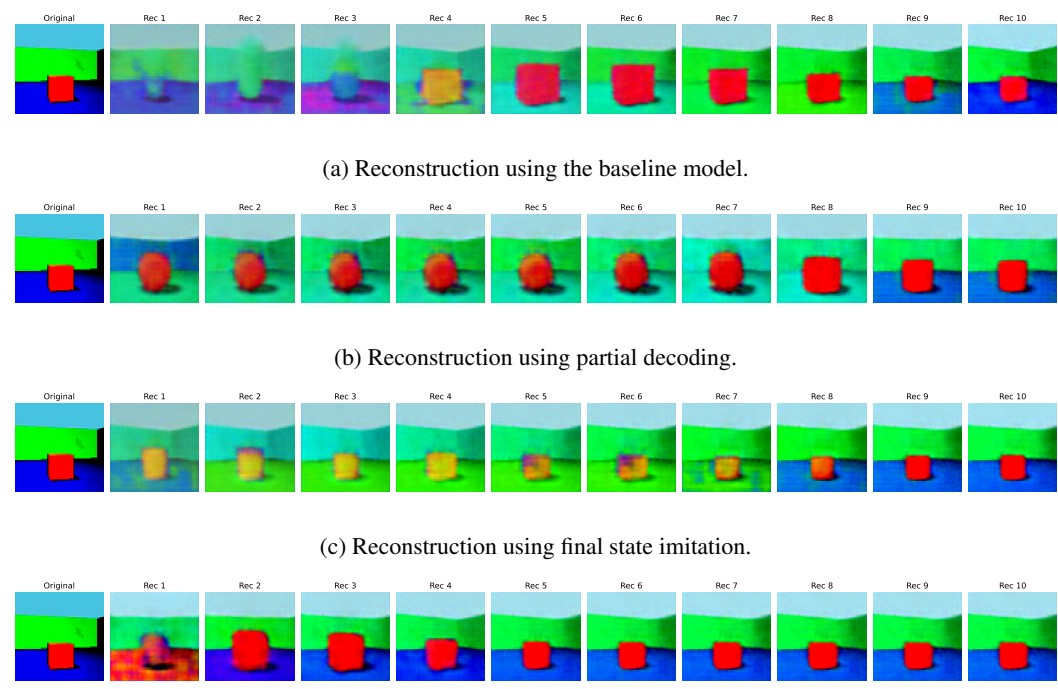

(a) Reconstruction using the baseline model.

(b) Reconstruction using partial decoding.

(c) Reconstruction using final state imitation.

(d) Reconstruction using PD+FiSI+PDM.

Figure 7: Reconstruction for the same example with different methods.

We study the four measures of variation presented in the paper, namely **synonymy**, **homonymy**, **word order freedom** and **entanglement**. **Synonymy** measures the presence of one-to-many mappings between atomic meanings and characters in a position, being minimized when each generating factor value is mapped to a single position and character. **Homonymy** measures the opposite, i.e., the presence of many-to-one mappings, and is minimized when each character in a position is mapped to a unique generating factor value. **Word order freedom** refers to the strictness of the mapping between generating factors and positions in the message (for example, if the mapping system always encodes the *shape* generating factor at the first position). It is minimized if all single generating factors are encoded in the same message position. Similarly, **entanglement** is minimized when all factors are encoded into unique positions in the message.

Important formular metricas, pero no encaja tan bien. Unico paper que conocemos. No clear patterns, but we argue it is because... An important assumption made by the authors for all these metrics is that meaning should be undivisably encoded into single positions of the message. We believe this scope of regularity is too narrow to capture compositionality-respecting mappings in out setting, as defined in F.2, and does not fit well when the space of messages is greatly larger than the amount of possible states. We found no clear trend or similar behavior across both tested datasets, save a small decrease in **entanglement** when using **PD**. This may suggest that the increased pressure for efficiency forces individual generating factors to be uniquely distributed in the message, however we believe there is not sufficient evidence to make such a claim considering the previously discussed pitfalls of the metrics.

To our knowledge there are no available official implementations for these metrics. Our versions can be found in the official repository for this work.

Table 4: Permutation tests for Shapes3D and MPI3D on emergent language metrics.

| Metric | Mode A | Mode B | Statistic | p-Value |
|---|---|---|---|---|
| TopSim | Progressive Decoding (PD; ours) | Full message reconstruction; no IL (Baseline) | 0.0263 | **0.0233** |
| TopSim | Full state imitation | Final-State Imitation (FiSI; ours) | -0.0267 | 0.1014 |
| TopSim | Full state imitation | Message imitation (Baseline) | -0.0018 | 0.9332 |
| TopSim | Final-State Imitation (FiSI; ours) | Message imitation (Baseline) | 0.0249 | 0.2605 |
| TopSim | PD+FiSI+KoLeo $\lambda = 1.5$ | PD+FiSI+PDM $\lambda = 1.5$ | -0.0371 | **0.0004** |
| Useful Length | Progressive Decoding (PD; ours) | Full message reconstruction; no IL (Baseline) | -2.5 | **0.0031** |
| Useful Length | Full state imitation | Final-State Imitation (FiSI; ours) | 0.0 | 1.0 |
| Useful Length | Full state imitation | Message imitation (Baseline) | 0.0 | 1.0 |
| Useful Length | Final-State Imitation (FiSI; ours) | Message imitation (Baseline) | 0.0 | 1.0 |
| Useful Length | PD+FiSI+KoLeo $\lambda = 1.5$ | PD+FiSI+PDM $\lambda = 1.5$ | 1.3000 | 0.0996 |
| Final MSE | Progressive Decoding (PD; ours) | Full message reconstruction; no IL (Baseline) | -0.0218 | 0.0657 |
| Final MSE | Full state imitation | Final-State Imitation (FiSI; ours) | -0.0130 | 0.4871 |
| Final MSE | Full state imitation | Message imitation (Baseline) | -0.0201 | 0.5679 |
| Final MSE | Final-State Imitation (FiSI; ours) | Message imitation (Baseline) | -0.0071 | 0.8384 |
| Final MSE | PD+FiSI+KoLeo $\lambda = 1.5$ | PD+FiSI+PDM $\lambda = 1.5$ | 0.0343 | **0.0431** |

(a) Shapes3D permutation test

| Metric | Mode A | Mode B | Statistic | p-Value |
|---|---|---|---|---|
| TopSim | Progressive Decoding (PD; ours) | Full message reconstruction; no IL (Baseline) | 0.0037 | 0.4416 |
| TopSim | Full state imitation | Final-State Imitation (FiSI; ours) | -0.0193 | **0.0019** |
| TopSim | Full state imitation | Message imitation (Baseline) | 0.0047 | 0.4395 |
| TopSim | Final-State Imitation (FiSI; ours) | Message imitation (Baseline) | 0.0240 | **0.0007** |
| TopSim | PD+FiSI+KoLeo $\lambda = 1.5$ | PD+FiSI+PDM $\lambda = 1.5$ | -0.0060 | 0.4767 |
| Useful Length | Progressive Decoding (PD; ours) | Full message reconstruction; no IL (Baseline) | -2.7000 | **0.0345** |
| Useful Length | Full state imitation | Final-State Imitation (FiSI; ours) | 0.0231 | 1.0 |
| Useful Length | Full state imitation | Message imitation (Baseline) | 0.2231 | 0.3998 |
| Useful Length | Final-State Imitation (FiSI; ours) | Message imitation (Baseline) | 0.2000 | 0.5820 |
| Useful Length | PD+FiSI+KoLeo $\lambda = 1.5$ | PD+FiSI+PDM $\lambda = 1.5$ | 0.2222 | 1.0 |
| Final MSE | Progressive Decoding (PD; ours) | Full message reconstruction; no IL (Baseline) | 0.0011 | 0.5894 |
| Final MSE | Full state imitation | Final-State Imitation (FiSI; ours) | -0.0018 | 0.1166 |
| Final MSE | Full state imitation | Message imitation (Baseline) | 0.0019 | 0.2870 |
| Final MSE | Final-State Imitation (FiSI; ours) | Message imitation (Baseline) | 0.0037 | **0.0201** |
| Final MSE | PD+FiSI+KoLeo $\lambda = 1.5$ | PD+FiSI+PDM $\lambda = 1.5$ | -0.000012 | 0.9626 |

(b) MPI3D permutation test

# J  Permutation Tests

## J.1  Emergent language metrics

To assess the statistical significance of each component added to the Iterated Learning (IL) framework proposed in this work, we conducted permutation tests using the *SciPy* library [67]. The test statistic was defined as the difference in means, $\bar{A} - \bar{B}$, where $\bar{A}$ corresponds to the arithmetic mean of the evaluated mode and metric across 10 random seeds. The complete results are presented in Table 5.

Across both the Shapes3D and MPI3D datasets, we found that the inclusion of the PD module consistently reduced the useful message length, yielding a test statistic of approximately $2.5$ and a $p$-value below $0.05$. None of the other additions introduced in this work produced a statistically significant decrease in useful length relative to their respective baselines, supporting our hypothesis that PD imposes an efficiency pressure on the emergent language. Moreover, we observed a statistically significant increase in TopSim for the Shapes3D dataset, with a test statistic of $0.026$ and $p < 0.05$.

Final-State Imitation had a statistically significant gain in TopSim over both baseline methods on MPI3D, with statistic of $\sim 0.02$ and $p < 0.02$. PDM outperformed the KoLeo estimator in both reconstruction error and TopSim on Shapes3D, but achieved no statistically significant increase in MPI3D.

## J.2  Disentanglement permutation tests

In this section we evaluate the statistical significance of the differences between our model and the discrete baseline on the metrics used in table 2. Similar to the previous section, we conducted

permutation tests, using the difference in means as a statistic for 10 random seeds. The complete results are presented in Table 5.

We found a significant difference in accuracy on the Shapes3D dataset, with a statistic of 0.114 and $p < 0.01$. We found a significant descrease in RMSE on both datasets, with statistics of 0.227 and 1.151 for Shapes3D and MPI3D, respectively, and $p \leq 0.001$. We also observed a significant gain in disentanglement on both datasets, with statistics of 0.003 and 0.023 for Shapes3D and MPI3D, respectively, and $p < 0.05$ for both.

Table 5: Permutation tests for Shapes3D and MPI3D on disentanglement, accuracy and RMSE using messages from discrete models as data for a linnear probe.

| Metric | Mode A | Mode B | Diff(A-B) | p-Value |
|---|---|---|---|---|
| Accuracy | VAE+EL | VAE+CELEBI | -0.1139 | **0.0018** |
| RMSE | VAE+EL | VAE+CELEBI | 0.2266 | **0.0010** |
| DCI Disentanglement | VAE+EL | VAE+CELEBI | -0.0034 | **0.0042** |

(a) Shapes3D permutation test

| Metric | Mode A | Mode B | Diff(A-B) | p-Value |
|---|---|---|---|---|
| Accuracy | VAE+EL | VAE+CELEBI | -0.0128 | 0.0570 |
| RMSE | VAE+EL | VAE+CELEBI | 1.1512 | **0.0010** |
| DCI Disentanglement | VAE+EL | VAE+CELEBI | -0.0230 | **0.0002** |

(b) MPI3D permutation test

