# OpenReview forum: "A Compressive-Expressive Communication Framework for Compositional Representations"
_NeurIPS.cc/2025/Conference — NeurIPS 2025 poster_

### Official Review · Reviewer_ct41 · 2025-06-13

**Clarity:** 3
**Significance:** 2
**Originality:** 2
**Rating:** 4
**Confidence:** 4

**Summary:**

The paper concerns the problem of learning more compositional representations in a discrete auto-encoder framework. For instance, given image data, how can we learn discrete latent representation that show more signs of compositionality? Their contribution is to propose three inductive biases for this, inspired by the cognitive science and AI literature on emergent communication:

1. **Progressive reconstruction**: The input should be partially reconstructable from individual sub-messages in the latent space, rather than only from the entire message.
2. **Iterative learning**: A student encoder distils the reconstruction behaviour from a previous generation’s frozen encoder/decoder, over the course of several generations. This sort of approach has been shown in prior work to increase the compositional structure of languages, and the authors’ version differs slightly in that the messages themselves are not imitated in subsequent generations (only the final output of the decoder).
3. **Message entropy maximization**: The messages across different encoded inputs should be different, so as to improve expressivity and prevent collapse.

The paper then presents evidence that these three inductive biases improve the compositionality of representations without significantly degrading input reconstruction performance. They do this by looking at compositionality metrics (Topological Similarity, ground-truth generative factor decoding, Disentanglement) across two simple synthetic datasets in which the ground-truth generative factors are known.

**Questions:**

1. The progressive reconstruction loss might encourage two things, and it’s not entirely clear which contributes to the results: (1) short messages, (2) messages in which each individual word carries information about the message (i.e., not holistic). Which do you think is driving the results? 2 seems more related to compositionality (in fact, as you say, 1 could harm compositionality by encouraging the use of a single position with a very large vocabulary!), but at the same time it seems like there could be other, better regularizers for 2 centered around mutual information that wouldn’t also encourage (1).
2. To what extent is the discretization important in any of this work? It’s meaningful to talk about compositionality w.r.t. continuous features, and all of your 3 proposed regularization methods would also work for a continuous latent space (or block-wise continuous space, as in the object-centric learning literature), would they not? I understand that you want to make a connection to the emergent language literature, but really your method is about learning compositional latent representations in an auto-encoder more generally, and nothing obviously requires the discrete aspect. At the very least, it would be interesting to know whether your inductive biases also contribute to compositional representations in fully continuous architectures.
3. You say that larger values of $\lambda$ can harm compositionality and reference Fig. 2. I don’t see that; in Fig. 2 TopSim just seems to plateau past 1 on both datasets. Can you clarify?
4. You say that part of the entropy-maximization regularizer is to avoid collapse during the imitation phase. Why would the imitation phase result in collapse if the previous generation’s sender and receiver, as well as the current generation’s receiver, are frozen? I can maybe see how this could happen with *many* generations, given that a collapsed message would be a fixed point of this iterative learning process, but (a) its not obvious that the iterative learning dynamics would lead to this fixed point (e.g., they could instead diverge or rotate), and (b) presumably that would take many generations. Can you please explain?
5. When discussing PDM results, you say that for MPI3D this setting achieves the highest TopSim but increases useful message length and reconstruction loss compared to PD alone. You then say that “this highlights a trade-off between expressivity and compressibility, in line with prior theoretical accounts on communication efficiency and compositionality”. I do not see how this result highlights this tradeoff. In particular, the pairwise distance maximization component has not further increased the structure or compressibility of the language (TopSim), nor has it increased the languages expressivity as demonstrated by the worse reconstruction performance. What do you mean, then?

**Ethical Concerns:**

["NO or VERY MINOR ethics concerns only"]

**Final Justification:**

Following the authors' rebuttal, I've chosen to maintain my score. The authors engaged with my feedback and have said that they will change the paper accordingly, but my main justification for not increasing my score is that the proposed regularizers and inductive biases do not increase the authors' compositionality score very much, and do harm task performance. The paper presents a nice set of ideas for inductive biases for compositionality, which I think is a research direction we should continue pursuing, but given that the effect of the inductive biases is not very string I think a score of "borderline accept" is fair.

**Limitations:**

Yes.

**Paper Formatting Concerns:**

None.

**Quality:**

3

**Strengths And Weaknesses:**

**Strengths**

1. I like that the paper is trying to design multiple inductive biases to effectively improve the compositionality of discrete auto-encoders. It’s an important problem, and I think others can continue to build on this work in the future.
2. For the most part, the paper is quite clearly written. It’s easy to get a sense of the motivation, mechanistically understand the methods, and interpret the experimental results. All of this should help broadcast the findings to the community and improve reproducibility.
3. Although I’m going to list a lot of weaknesses and questions below, my overall impression of the paper is net positive as a result of the two points above: the motivation is good, the problem is important, and the paper is easy to digest.

**Weaknesses**

1. **Suggestion**: The emergent communication, expressivity/compressibility, and iterative learning literature from cognitive science and AI are significant starting points for this work. For people already familiar with the literature, the paper’s motivation and the components of the model are easy to follow and their justification is clear. For others unfamiliar with this line of work, I worry that the motivation and model components will seem obscure and ad-hoc. I think early on in the paper you can say a bit more about what this prior literature has found, and supplement it with some concrete examples of languages that emerge with and without pressures for compression like iterative learning.
2. **Suggestion**: The progressive decoding part of the model is partly motivated in the text by chain-of-thought reasoning. I think the link here between progressive decoding, compositionality, and chain-of-thought reasoning is actually rather speculative and not entirely justified (a particular step in a chain-of-thought might not help solve the problem at all in isolation, and might only help in the context of other thoughts in the chain). Instead, a more direct link can be made to the systematic structure this progressive decoding induces in the message-to-image mapping. In particular, it makes the function simpler because each individual word must contribute to the meaning, which eliminates holistic mappings as a solution. It’s fairly easy for instance to see how progressive decoding can directly lead to higher topological similarity (your metric for compositionality), so you may as well motivate progressive decoding through this direct link to structured mappings and topological similarity in the first place.
3. There’s a clear link to object-centric learning, in particular slot-type of architectures that have additive decoders, especially with regards to the progressive decoding component of the model. This should be discussed in the related work, in addition to the discussion on beta VAEs.
4. The methods are grounded in emergent communication games, but I feel like they can go beyond that. For instance, such an encoder/decoder can be used in an auxiliary loss that places inductive bias for compositionality on another neural network’s representations for arbitrary tasks (i.e., train a neural network to solve some arbitrary task, consider some latent representation in this neural network as the “image” in the current framework, and then alternative training your encoder/decoder’s weights w.r.t. the losses in the current paper vs. training the representation w.r.t. its task loss and the losses in the current paper). Hopefully I’m being clear enough here, but I think such an approach could be quite interesting. I’m not saying this has to be done in the current work of course, but saying some more about how you think your work could be useful for representation learning in general (and not just emergent communication) would be helpful in broadening the scope of the ideas.
5. There’s a lot of mathematical obfuscation in the section introducing FiSI that I don’t think is actually necessary, and at best can go in the Appendix. In particular, it seems completely clear that only imitating the final output of the previous iteration’s receiver provides more flexibility to the sender to construct different sorts of messages, and we don’t need to go through a tedious mathematical argument involving significant notation and a proposition in order to understand this. In my opinion this breaks the flow of the paper.
6. Overall, one could argue that the improvements in compositionality (as measured by TopSim in Table 1) are not actually that significant. For instance, final state imitation yields only very marginal improvements over message imitation on both datasets, and these marginal improvements actually fall within the standard error. The results therefore don’t clearly support the proposed inductive biases as much as you seem to suggest, and I would be very curious to hear your rebuttal on this point — perhaps I’m missing something.

---

> ### Author Rebuttal · Authors · 2025-07-31
>
> ### Rebuttal Reviewer ct41
>
> We thank the reviewer for their deep and insightful response and valuable suggestions, which we believe will significantly improve the paper.
>
> ---
>
> ### Weaknesses / Suggestions
>
> 1. We appreciate the suggestion to expand the introduction and related work to provide readers with necessary background on the model components and pressures in the iterated learning setup, which our strategy challenges. We agree this will help clarify the paper's positioning—thank you again for the recommendation!
>
> 2. This suggestion has two parts:
>    1. We agree that the link to chain-of-thought (CoT) is speculative and partial. The only shared property is that CoT reasoning steps, like our message tokens, exhibit compositional and sequential structure. However, CoT subsequences are not explicitly pressured to lead to a correct conclusion/output, making the analogy misleading. We plan to remove this comparison.
>    2. Thank you for suggesting we clarify the connection between progressive decoding and compositional metrics like topographic similarity. While we mention this in line 127, we will add a clarifying sentence in the final version.
>
> 3. Thank you for highlighting the connection to object-centric learning, slot-type architectures, and additive decoders. We will add a discussion of these in the related work section.
>
> 4. We appreciate the suggestion to broaden the scope beyond emergent communication. We are indeed exploring further uses of our signaling architectures for representation learning, and the idea of cross-model communication is promising if fine-tuned appropriately. We will add a discussion of this in the "Limitations and Future Work" section.
>
> 5. We agree with the suggestion to move Proposition 3.1 to the appendix and will do so in the final version.
>
> 6. **Regarding Table 1**:
>    To assess statistical significance, we added 5 seeds per method and ran permutation tests, confirming that the impact of our components on Topsim is statistically significant. The updated table and p-values are in the rebuttal to reviewer Likm; results for Table 2 are in the rebuttal to reviewer rCnw.
>    Concerning inductive biases, we note that gains in Topsim are stronger when combining PD+FiSI+PDM, compared to applying them separately. That the components are less effective individually does not contradict our claims. On the contrary, it supports the motivation for combining them—each pressure alone may be insufficient, but together they produce synergy.
>    For example, FiSI is meant to allow sender strategy drift, but this drift doesn't arise naturally from optimization. Hence, we introduce PDM to explicitly encourage such exploration. We will clarify this in the final version.
>
> ---
>
> ### Questions
>
> 1. **Progressive reconstruction loss**:
>    1. This loss clearly enforces (1)—penalizing longer messages—since later tokens contribute more to the loss. As for (2)—penalizing tokens that carry no information—we do not claim this. Instead, we argue it promotes interpretability in *submessages* (prefixes of increasing length), as the receiver is penalized for failure to reconstruct from partial messages.
>    2. Encouraging shorter messages aligns with a core pressure in Iterated Learning theory. Thus, we prefer regularizers that enforce both compression and interpretability (e.g., progressive loss) over those that only target informativeness. Efficient encoding is a necessary (though not sufficient) condition for low Kolmogorov complexity, which underlies emergent compositionality. Appendix F explains why our regularization is needed, as training data covers only a small portion of possible factor combinations.
>
> 2. **Use with continuous features**:
>    In principle, our setup could be extended to continuous features. However, in practice, discrete and separated encodings have proven more effective for compositionality in iterated learning (see [45]). Discreteness mirrors human vocabulary and enables concept separation.
>    Technically, discreteness supports a richer encoding space and helps avoid mode collapse, as shown in VQ-VAE literature. We believe this is especially important in combinatorially extended message spaces like ours, where collapse risks are amplified. We will emphasize this point in the related work section, linking it to object-centric learning.
>
> 3. **Clarification on λ and TopSim**:
>    Thank you for catching this. We previously wrote (lines 238–239) that “larger values of λ encourage more informative messages, but reduce alignment with generating factors,” to justify the claim that “intermediate values of λ are best.” As you noted, this isn't clearly supported by the data. What does drive the optimal λ value is the emergence of a plateau in TopSim, while useful length continues to increase. We will correct this explanation.
>
> 4. **Clarification on “collapse”**:
>    In hindsight, our use of “collapse” was poorly worded. We appreciate the reviewer pointing this out. What we meant was that, without additional pressure, a new sender has no incentive to drift from the previous sender’s trajectory—i.e., the decoding sequence from submessages of increasing length remains unchanged.
>    By adding entropy regularization, we encourage exploration of alternative encoding strategies, similar to how entropy is used to promote exploration in reinforcement learning. The actual “collapse” occurs over generations. We will revise the wording accordingly.
>
> 5. **Trade-off between expressivity and compressibility**:
>    By this, we mean that adding PDM improves TopSim (from 0.141 to 0.16), but the other two metrics worsen (see second vs. last row in the second part of Table 1). This contrasts with the reviewer’s statement that PDM “has not further increased the structure or comprehensibility of the language (TopSim),” so we may be misunderstanding their comment. Did we miss something?

---

> > ### Comment · Reviewer_ct41 · 2025-08-01
> >
> > Thank you for taking my feedback into consideration and answering my questions. I think I'm still comfortable with my score, owing mainly to the very small (even if statistically significant) improvements in compositionality from the proposed method (at the cost of worse task performance, often).
> >
> > Regarding my final question and your reply, I was specifically talking about using PDM compared to KoLeo, in which case the topological similarity scores on MPI3D are identical. I've seen now though that in the text you acknowledge this at the end of Section 4.5, so that was something I missed on my part.

---

### Official Review · Reviewer_gfpt · 2025-07-02

**Clarity:** 3
**Significance:** 2
**Originality:** 2
**Rating:** 4
**Confidence:** 1

**Summary:**

The authors study the notion of compositional generalization through a communication framework - the task for a receiver is to decode the sender's image based on finite-space messages provided from the sender. These messages define the framework for compositional generalization in language in the sense that a compositional language for sending messages can lead to more expressive and also more compressive signaling scheme. The authors propose three additions to the existing scheme to allow for more informative (and hence more compositional) messages in the signaling phase: progressive decoding to provide intermediate signal which can lead to providing more informative and useful signals earlier on, a final-state imitation procedure that provides informative signal for the full message decoding, and pairwise distance maximization which incentivizes diversity in messages. Their overall scheme leads to improved performance in terms of reconstruction metrics as well as the length of messages required.

**Questions:**

- Could the authors provide further clarification on the use of Equation (3)? How does this contrast to progressive decoding since it also has a term that corresponds to final reconstruction after full message?

**Ethical Concerns:**

["NO or VERY MINOR ethics concerns only"]

**Final Justification:**

The authors propose a framework for studying and improving emergent communication schemes through progressive decoding, final-state imitation and use of a diversity regularizer. Since the topic of research in this work is outside my area of expertise, I have understood the high-level motivation behind each of the proposed contribution but am not familiar with the details - this is reflected in my score and confidence.

**Limitations:**

No further limitations than what is discussed above.

**Paper Formatting Concerns:**

No concerns.

**Quality:**

3

**Strengths And Weaknesses:**

**Strengths**

- The paper is clearly written and tackles an interesting and relevant problem.
- The use of progressive decoding is quite intuitive since in language, we do want to maintain that earlier symbols make the future predictable which is the underlying objective here.
- The evaluation clearly shows improved performance on both output space similarity as well as length of sentences.

**Weaknesses**

- Current evaluation is solely on synthetic problems and it is not clear how this framework extends to problems that can have more complex objects as well as relations between the objects.
- An important question not being directly tackled here is the case where the language of communication is English; how does this framework change and behave when we rely on such a language? A core reason for compositional generalization is the fact that the same language is useful for a wide variety of tasks, and such an analysis is missing from the current experiments.
- While the authors claim to tackle compositional generalization, they aren't really evaluating for this from the metrics. Have the authors considered multi-object scenes where evaluation involves asking a pre-trained model whether certain relations are respected or not in reconstruction?

---

> ### Author Rebuttal · Authors · 2025-07-31
>
> ### Rebuttal Reviewer gfpt
>
> We thank the reviewer for their time and thoughtful response.
>
> The reviewer's concerns primarily highlight the need for evaluation on more realistic or complex datasets, or the use of English for communication. This may reflect a misunderstanding of our paper’s main positioning, which focuses on emergent language via iterated learning. In this setting, controlled scenarios are useful in multiple ways, and using English as a medium of emergent communication is not straightforward. We expand on this below.
>
> ---
>
> ### Weaknesses
>
> 1. **Evaluation via non-synthetic problems**:
>    Thank you for raising this important point.
>
>    We acknowledge that our experiments are conducted on synthetic datasets. This is intentional and *aligned with common practice* in the emergent communication and compositional generalization literature. Controlled environments are essential for systematically studying inductive biases and the emergence of structured communication protocols (indeed, nearly all cited references on emergent language use synthetic datasets of comparable complexity—e.g., [5], [12], [14], [23], [24], [35], [37], [38], [44], [47], [48], [52], [58], [64]).
>
>    Specifically, synthetic datasets provide several benefits:
>    1. **Controlled data distributions**: They allow precise control over training and test distributions—especially important for implementing compositional splits that test generalization to novel attribute combinations, which is central to our study.
>    2. **Ground-truth factorization**: Meaningful evaluation of compositionality requires access to generative factors (e.g., shape, color, size). Real-world datasets often lack this disentangled ground truth, making metrics like topographic similarity difficult or ambiguous to apply.
>    3. **Complexity minimization**: Following Occam’s razor, we avoid unnecessary complexity to preserve clarity in causal interpretation. Introducing relational reasoning or complex object structures would increase dimensionality and introduce new, confounding effects that could obscure the impact of our proposed mechanisms.
>
>    While evaluating more complex and realistic tasks is a valuable future direction (as noted below), such extensions introduce orthogonal challenges. We believe our current setup provides a principled and interpretable testbed for isolating the effects of our mechanisms on compositionality. We’ve added a note about this extension path in the discussion section.
>
> 2. **English as a communication medium**:
>    We appreciate the reviewer’s thoughtful comment. However, this concern seems to arise from a misunderstanding of our objective.
>
>    Our framework does not aim to model or analyze evolved human languages such as English, but instead studies the emergence of **compositional communication protocols de novo** between artificial agents (see lines 44–47). This is motivated by foundational questions in cognitive science [29], where a key challenge is understanding how compositional structures arise from basic communicative dynamics and learning mechanisms.
>
>    A core theoretical motivation in studying emergent language is to test whether compositionality emerges naturally from optimizing for efficient and generalizable communication, as proposed by recent information-theoretic accounts [43]. Thus, our work should be viewed in the context of **emergent communication and compositional learning**, not natural language modeling.
>
>    While English is indeed compositional, it is the product of centuries of cultural evolution shaped by complex social and historical factors—many of which are not tractable in computational models. Emergent languages, in contrast, enable **precise control and isolation of contributing factors**, allowing systematic analysis of their role in shaping communication.
>
> 3. **Multi-object scenes and relational reasoning**:
>    1. We respectfully disagree with the claim that our work does not evaluate compositional generalization. As noted in line 212, we adopt the compositional generalization split introduced by [52], which is specifically designed to test generalization to novel combinations of known factors—a core benchmark for compositionality. We apply this to two well-established datasets, Shapes3D and MPI3D, widely used for analyzing compositional structure in learned representations.
>    2. Evaluating multi-object scenes with relational queries is indeed an exciting direction, but we believe it is outside the scope of our current study. Our goal is to isolate and understand how our mechanisms affect emergent message structure and efficiency. Adding relational reasoning would introduce additional complexity and confounds, making attribution to our contributions less clear.
>    3. We focus intentionally on the simplest form of compositionality: generalization over novel combinations of independent latent factors (e.g., shape, color, size). We do not address more complex compositional structures (e.g., syntax or multi-object relational reasoning), which introduce additional challenges (see 1.iii above).
>
>    Nonetheless, we appreciate the reviewer’s suggestion and plan to explore relational settings in future work.
>
> ---
>
> ### References
> [29] Simon Kirby et al., “Compression and communication in the cultural evolution of linguistic structure,” *Cognition*, 2015.
> [43] Yi Ren et al., “Compositional languages emerge in a neural iterated learning model,” 2020.
> [52] Lukas Schott et al., “Visual representation learning does not generalize strongly within the same domain,” ICLR, 2022.
>
> ---
>
> ### Questions
>
> To our understanding, the reviewer asks how Eq. (3) contrasts with Eq. (1) used for progressive decoding (PD). The key distinction is that Eq. (3) applies to the **imitation phase**, whereas Eq. (1) pertains to the **interaction phase**, which has significant implications:
>
> 1. In the interaction phase (Eq. 1), both sender and receiver are unfrozen, and the loss measures reconstruction accuracy from input to output. In the imitation phase (Eq. 3), the receiver is frozen, and the sender imitates the *reconstruction* produced by the previous-generation sender, not the ground truth (see line 172).
> 2. The left and right components of Eq. (3) compare full vs. final reconstruction. Our choice—to use only final state reconstruction during imitation (as in our Eq. 2, corresponding to the right-hand side of Eq. 3)—supports broader sender exploration when combined with PDM (see §3.2.2). This differs fundamentally from PD, where sender and receiver jointly optimize under an efficiency constraint (minimizing useful message length) imposed by Eq. (1).
>
> We hope this clears up the confusion—please let us know if we can clarify further.

---

> ### Comment · Area_Chair_2oGp · 2025-08-04
> **AC Request**
>
> Dear Reviewer,
>
> Thank you very much for taking the time to review this paper.
> Would you kindly confirm whether the authors' response adequately addresses the third concern you raised under "Weaknesses"?
>
> Best regards,
> AC

---

> > ### Comment · Reviewer_gfpt · 2025-08-05
> > **Reviewer Response**
> >
> > Thanks to the authors for providing clarifications. I have updated my rating accordingly.

---

### Official Review · Reviewer_rCnw · 2025-07-02

**Clarity:** 2
**Significance:** 2
**Originality:** 2
**Rating:** 4
**Confidence:** 3

**Summary:**

This paper investigated compositional generalisation in an emergent communication paradigm through a newly introduced framework (CELEBI). This framework comprises of three mechanisms: interactive decoding, reconstruction-based imitation learning, and pairwise distance maximisation. In a series of experiments and ablations, the papers claims that the emerged languages are more efficient and compositional as opposed to standard (baseline) emergent communication methods on two image-based datasets. The paper incorporated many experiments and derivations that make it a complete paper, but I found it somewhat difficult to assess exactly what the main position of this paper was – emergent communication or compositional generalisation. I did not completely check the correctness of the mathematical explanations and proposition in the main paper and appendix, I adjusted my confidence accordingly.

**Questions:**

1. As mentioned in points 1,2 and 3, I am not convinced of the effectiveness of the (individual) proposed mechanisms. Could the authors provide information regarding the test for significance and the resulting statistics? If this is available, adding it to table 1 and 2 would be appreciated.
2. Some qualitative interpretation of the emerged languages would be very valuable and strengthen the claims of this paper. How exactly do the messages change as a result of the individual mechanisms? Here, I would be especially interested in the changes resulting from PDM. In addition to manually looking for message meanings, directed pertubations of the messages could give insight into how the individual elements of messages alter the receivers’ reconstruction. This would moreover strengthen the relation between the datasets’ attributes and the actual image features (point 4 of the weaknesses). Although not explicitly mentioned, I assume that TopSim is calculated using the values of factors G instead of the pixel values, is this correct?
3. Figure 2a. Could the authors explain why the message useful length does not decrease as a result of higher lambda? Please correct me if I am wrong, but it seems to me that the introduced pressure (line 234) must also result in shorter useful length since a wrong prediction should more heavily incentive the speaker to use shorter messages.

**Ethical Concerns:**

["NO or VERY MINOR ethics concerns only"]

**Final Justification:**

Convinced by additional simulations, including statistical analyses and an additional metric for compositionality, presented by the authors during the rebuttal period, I increased my score with 1 point.

**Limitations:**

Yes

**Paper Formatting Concerns:**

-

**Quality:**

3

**Strengths And Weaknesses:**

As written in the summary, this is very elaborate work that incorporates methods and insights from multiple fields, making it an interesting contribution. Please see my assessment below.

Strengths:
1. Compositional generalisation is relevant in many areas concerning deep learning. This is also the case in emergent communication and has been a topic of interest for quite some time. This work is informed by findings in human language evolution, making comparisons with humans possible. Yet, this is not explicitly done, the authors draw on many relevant previous findings in emergent communication but fail to exactly show how this work relates to those findings. Nevertheless, I think this work is a step in the right direction.
2. I believe that the main contribution of this paper is the introduction of final-state imitation. Using the receivers’ interpretation to calculate an imitation loss is original and brings the simulation somewhat closer to human experiments with iterated learning in which a listener becomes the teacher of the next generation of listeners. It is however not clear how many generations are simulated.
3. The introduction of the ‘useful length’ metric is interesting. It allows the introduction of the geometric penalty (which seems to be undefined), and thereby shows that artificial agents, similar to humans, also show a trade-off between efficiency and expressivity.
4. The quality of this paper is good, it is fairly easy to follow, despite containing a considerable number of derivations which, in my view, are not necessary to make the point.

Weaknesses:
1. My main concern with this paper is that the results are somewhat marginal. Although the limitations section states that the results are significant (line 331, for which no proof is provided), the results in table 1 seem to be rather small. Some of the means and error margins overlap between conditions. For example, the authors state that final state imitation enhances compositionality and outperforms the non-IL baseline and surpasses message-level imitation (lines 248 – 250) but no statistical tests are provided. It would be appreciated if the authors could provide evidence regarding the significance of the results. Assuming that the results are significant, it is difficult to assess  from only TopSim whether the proposed methods advance compositional generalisation or teaches something about human language evolution. A clearer framing on what this paper is about would help. In case of the latter, it would be necessary to show some of the learned languages and try to interpret them (even though this is difficult). In case of the former, I would like to read some discussion as to why the proposed mechanisms are meaningful for non-emergent communication generalisation / how they extrapolate beyond emergent communication.
2. The results regarding length of useful messages in the progressive decoding objective are not necessarily surprising since the method, please correct me if I am wrong, essentially enforces the listeners to make a prediction early (line125, 126). What do we learn from this? Moreover, Table 2 does not convince me that the reconstruction error is improved resulting from PD.
3. Similar to the previous point, it seems that the results of table 2 are somewhat oversold in section 4.6. For example: in the case of the Shapes3D dataset, an increase of 0.002 with an error margin of 0.001 does not seem to be substantial nor significant. It is a bit more problematic since RMSE and the accuracy are worse for CELEBI. While it is difficult to unravel the underlying meanings of messages in emergent communication, these results do not inform us much of how the proposed CELEBI framework is different from other work. Some qualitative inspection would help. Similarly, if the focus of this paper is generalisation, then it seems to me, but I am willing to be convinced otherwise, that a mechanism is only advancing generalisation if the RMSE and accuracy would also improve. Could the authors elaborate on this?
4. Message stucture is currently only measured through TopSim. While it is indeed widely used in emergent communication literature, this metric also has limitations  [1, 2, 3]. In combination with images, it is unclear whether the agents actually use image features to create their messages [1] and it is argued that TopSim is not appropriate [2]. Moreover, TopSim does not capture compositionality when languages show variability [3]. Instead of relying on TopSim only, I would suggest to expand the set of metrics to include homonymy, synonymy, and word-order freedom as proposed by [3]. In addition to this, it would be helpful if the authors could qualitatively show that messages are indeed more structured when, e.g., pairwise distance maximisation is used (section 4.5).
5. Minor weakness: While the related work section suggests that progressive decoding is different from LazImpa, I find this somewhat misleading. Variable message lengths should not meaningfully change the effect of a pressure to make early predictions. Moreover, the PD objective also seems to implicitly be a length penalty (line 125, 126).

[1] Kouwenhoven et al. 2024. The curious case of representational alignment: Unravelling visio-linguistic tasks in emergent communication
[2] Chaabouni et al. 2022., Emergent Communication at Scale.
[3] Conklin et al. 2023. Compositionality with Variation Reliably Emerges Between Neural Networks

Minor issues:
1. It would be easier to understand the figures and tables if their captions would be more informative. The table captions could include the take-away message from that table, and incorporate the meanings of bold values / show significance values. The figures would similarly benefit: figure 1a is very clear, but how exactly should I interpret 1b? Are these generations, or predictions after each message component? Figure 2 similarly only discusses figure b.
2. Subsections 2.1 and 2.2 could be merged into one subsection to reduce space. They seem to explain the same thing.
3. Lines 137 – 139: iterated learning does not *force* languages to be easy to learn, the languages resulting from IL are easier to learn because the elements that are easy to learn / compress are more likely to be transmitted. The end result of several generations of IL is that the languages are thus easier to learn.
4. Line 156: during THE imitation phase
5. Line 202: perhaps you could change the section name to “Methods” and start the result section at what is currently subsection 4.3.
6. Line 266: decrease?

---

> ### Author Rebuttal · Authors · 2025-07-31
>
> ### Rebuttal Reviewer rCnw
>
> We thank the reviewer for their time and thoughtful response.
>
> The reviewer asks whether our paper is primarily about emergent communication or compositional generalization. In hindsight, we recognize the abstract may have caused confusion. Our main focus is emergent communication, and we will clarify this in the revised abstract.
>
> ### Weaknesses
>
> 1. **Response to two concerns**: (reference numbers [1], [2], [3] as in the reviewer's text, [64] as in our paper's draft)
>    1. **Table 1 and lines 331, 248–250**: The reviewer rightly notes the original Table 1 showed marginal improvements. We reran all methods with 5 additional seeds to increase statistical power and performed permutation tests across interaction, imitation, and regularization methods (see response to reviewer Likm). We’ll include these in the supplementary materials. The updated table shows a clearer trend—final state imitation generally outperforms message imitation—but statistical significance was only observed in MPI3D. We will clarify this.
>    2. **Positioning beyond emergent communication**: The main insight is that pressures for signal complexity (useful length), expressivity (reconstruction error), and compositionality (topsim) arise from different aspects of the interaction setup and must be balanced for more compositional representations. This underlies our three innovations. While generalizing beyond emergent communication is a broader debate (e.g., [1]), our approach resembles complexity regularization methods like self-distillation [2]. We will elaborate on this in the "Limitations and Future Work" section.
>
> 2. **Progressive Decoding nontriviality**: PD acts as a length penalty, but the trade-off it induces with reconstruction accuracy is nuanced. As shown in Figure 3 (App. D), PD can prioritize efficiency (shorter useful length) over accuracy. The tuning of λ becomes critical. Partial decoding does not necessarily improve reconstruction but enhances efficiency. The key challenge is balancing concise and expressive messages while avoiding ambiguity.
>
> 3. **Table 2 – two clarifications**:
>    1. **RMSE metric**: Table 2’s RMSE does not reflect reconstruction error but evaluates representations on a downstream regression task, following [64]. This differs from true reconstruction performance and is less directly tied to compositionality and interpretability (similar to discriminative metrics in 4.iii). We'll clarify this distinction and its implications in the final version, as also noted by reviewer Likm.
>    2. **Accuracy and significance**: We increased each method to 10 runs (see updated table) and ran permutation tests comparing CELEBI to the EL baseline. Relevant comparisons: In Shapes3D, CELEBI outperforms EL in accuracy (p=0.0017), RMSE (p=0.001), and disentanglement (p=0.0041). In MPI3D, CELEBI exceeds EL in RMSE (p=0.0009) and disentanglement (p=0.0001). Full results will be in the paper.
>
> 4. **Multiple points**:
>    1. **Interpretability metrics**: Following the suggestion, we added synonymy, homonymy, freedom, and disentanglement (equal to "1 - entanglement" from Conklin et al. [1]) to the supplementary. Permutation tests show PD and PDM reduce entanglement. In Shapes3D, PD outperforms the interaction baseline (p=0.00059, diff=0.041); in MPI3D (p=0.0425, diff=0.064). PDM beats KoLeo (p=0.0052, diff=0.0195). Since reducing entanglement requires consistent encoding of different meanings, this supports PD's role in producing more compositional representations. We’ll expand on this in the final version.
>    2. **Qualitative results**: Besides the above metrics which we will add in a table, Appendix I contains generated examples showing regularities. We’ll add comparisons across methods to highlight improvements.
>    3. **Representation alignment from [3]**: In contrast to [3], where representations were not explicitly aligned with visual features, our setup uses a reconstruction task. Since agents must reconstruct images, the emergent language aligns naturally with underlying visual features. Topsim emerges from the correlation between generative factors and pixels—not class labels—ensuring grounded representations.
>    4. **Message structure**: Intermediate reconstructions often show meaningful transitions (App. I), unlike denoising noise. We'll add more examples to strengthen this point.
>
> 5. **Comparison with LazImpa**: Our PD loss is conceptually simpler and more interpretable. LazImpa's penalties depend on model-specific accuracy predictions, whereas ours apply a single global term (eq. 1), creating a clearer, system-wide efficiency pressure. Unlike LazImpa, where message length pressure is split across roles, PD treats the system as a whole. We’ll clarify this conceptual distinction in the revised version.
>
> ---
>
> ### Minor Issues
> 1. We will add takeaways in figure captions.
> 2. We will merge overlapping background sections.
> 3. We’ll revise to: "...favoring easy to learn languages..."
> 4. Thank you!
> 5. We will use "experimental setup" instead of "methodology."
> 6. The word “increase” refers to worse reconstruction, which we believe is correct.
>
> ---
>
> ### Questions
> 1. We have run permutation tests (see reviewer Likm) and will include them in the supplementary material.
> 2. Topsim is computed using generating factors. We will clarify this. We’ll also add qualitative examples showing the effect of changing message symbols on reconstruction. Without PDF support, we’ll reference these in Appendix I and describe the new figures.
> 3. **Effect of λ**: As λ increases, the PD loss asymptotically resembles full reconstruction, since the highest-order term dominates. In practice, λ ≈ 2.5 already saturates useful length. We’ll extend Figure 2 to λ=10 and explain this in §3.1.
>
> ---
>
> ### Table: Disentanglement Metrics for Shapes3D and MPI3D
>
> | Dataset  | Model        | Disentanglement ↑ | RMSE ↓        | Acc ↑          |
> |----------|--------------|-------------------|---------------|----------------|
> | Shapes3D | β-VAE        | 0.045 ± 0.004     | 2.460 ± 0.144 | 0.702 ± 0.025  |
> |          | β-TCVAE      | 0.043 ± 0.005     | **2.378 ± 0.123** | **0.721 ± 0.028** |
> |          | VAE + EL     | 0.108 ± 0.000     | 3.168 ± 0.027 | 0.343 ± 0.019  |
> |          | VAE + CELEBI | **0.112 ± 0.001** | 2.932 ± 0.045 | 0.453 ± 0.024  |
> | MPI3D    | β-VAE        | 0.031 ± 0.001     | 21.819 ± 0.111 | **0.509 ± 0.006** |
> |          | β-TCVAE      | 0.031 ± 0.001     | 21.854 ± 0.188 | 0.582 ± 0.005  |
> |          | VAE + EL     | 0.114 ± 0.002     | 15.971 ± 0.301 | 0.529 ± 0.006  |
> |          | VAE + CELEBI | **0.137 ± 0.002** | **14.82 ± 0.012** | 0.542 ± 0.002  |

---

> > ### Comment · Reviewer_rCnw · 2025-08-04
> >
> > I would like to thank the reviewers for their elaborate answers. I appreciate the additional simulations including statistical analyses and the additional metrics for compositionality. Perhaps it is worth adressing these new metrics in combination with the proposed qualitative examples. I updated my score.

---

> > > ### Author Response · Authors · 2025-08-05
> > >
> > > We thank the reviewer for their attentive response. We will address these new metrics more thoroughly along with the qualitative examples in appendix I to make a coherent discussion.

---

### Official Review · Reviewer_Likm · 2025-07-06

**Clarity:** 2
**Significance:** 2
**Originality:** 3
**Rating:** 4
**Confidence:** 2

**Summary:**

This paper proposed CELEBI, a method for learning compositional representations that combines interactive decoding (partial reconstruction from intermediate representations)  and an imitation (similar to iterated learning). This method is evaluated on two datasets, Shapes3D and MPI3D, demonstrating its strengths compared to previous methods for constructing discrete representations.

**Questions:**

- I find the notation quite confusing, especially for Eq. 0 (the unnumbered display equation at the top of p3, right after line 95). Does GenX take in the set $\mathcal{G}$ or just $G$ (as in line 71)? What is the input to $S$, $\mathcal{D}$ or $\mathcal{X}$? And what is the output of $R$? The textual description states that the reconstruction is an image $x$ but this equation shows the set $\mathcal{X}$.
- How is $C$ determined? The notation implies it's fixed/constant, but the text suggests that it's variable.
- Tuning the efficiency pressure (lines 132-135): I'm a bit confused by this explanation. How do small values yield fast reconstructions? The smallest value is $\lambda=1$ and that would give equal weight to all sub-messages. Similarly, this section states that higher values of $\lambda$ yield longer messages, but based on Eq.1 it seems that for large $\lambda$ there's a much larger penalty for longer messages.

**Ethical Concerns:**

["NO or VERY MINOR ethics concerns only"]

**Final Justification:**

I still have some of the reservations I mentioned in regards to the overall significance of the results, but given the authors responsive rebuttal I feel a bit more positively about this work and am inclined to increase a my rating.

**Limitations:**

yes

**Quality:**

3

**Strengths And Weaknesses:**

**Strengths**
- This paper addresses an important question (i.e., how to learn compositional representations) by proposing a new method that combines existing techniques in a novel way. As such, it present an original work that could potentially be interesting and valuable for the NeurIPS community.
- The chosen datasets, experiments, and baselines seem adequate for evaluating the method.

**Weaknesses**
- The results seem somewhat incremental, making it hard to assess the significance of this work. That is, the quantitative results suggests only slight improvement compared to previous methods, and in particular, it seems that the proposed method does not perform quite well in terms of accuracy (Table 2). Perhaps even more importantly, it's unclear what is the nature of the learned compositional structure and what new insight it affords. I could see value in a new method that provides compositional representations that are more insightful and interpretable than prior methods, even if that method doesn't achieve the best possible accuracy, but I don't think this was demonstrated in the paper. I therefore think that the paper could benefit from a deeper analysis of the learned compositional structure.
- The paper could also benefit from more clarity in the presentation. For example, the notation is not always clear (see questions below), and the baselines are not sufficiently explained.

---

> ### Author Rebuttal · Authors · 2025-07-31
>
> ### Rebuttal Reviewer Likm
>
> We thank the reviewer for their time and their attentive response.
>
> We believe that a misunderstanding regarding the positioning of our paper underlies several of the reviewer’s concerns. Our focus is on proposing a new iterated learning framework in which emergent representations become compositional. We do not aim to resolve the ongoing debate around the relationship or trade-off between compositionality/disentanglement and downstream accuracy. This is especially relevant for the interpretation of Table 2 (see discussion below). The three main new components of our framew...
>
> ---
>
> ### Weaknesses
>
> 1. **Regarding Table 1**:
>    While the reviewer did not mention this table explicitly, we believe their remarks about “only slight improvement” may refer to it. Several reviewers pointed out that results in Table 1 may not be statistically significant.
>    To address this:
>    - We increased the number of runs per method to 10 (updated tables below).
>    - We conducted permutation tests across methods in interaction, imitation, and regularization.
>    Results indicate statistically significant differences:
>    - **Shapes3D**: Progressive decoding outperforms the baseline in Topsim (p = 0.0233) and useful length (p = 0.0031). PDM outperforms KoLeo in Topsim (p = 0.0004) and reconstruction error (p = 0.0431).
>    - **MPI3D**: FiSI outperforms message imitation in Topsim (p = 0.0007) and reconstruction error (p = 0.0201), and full state imitation in Topsim (p = 0.0019). PD outperforms full message reconstruction in useful length (p = 0.0201).
>    Full results will be included in the final paper. Let us know if further comparisons are needed.
>
> 2. **Regarding Table 2**:
>    As noted by the reviewer, our method improves disentanglement but underperforms in accuracy relative to continuous VAEs, while slightly outperforming the discrete baseline. These outcomes align with literature (see [1]): disentanglement often harms accuracy in downstream classification.
>    Discrete representations may be less accessible to simple classifiers (as shown in [1]), and inductive biases for compositionality do not imply disentanglement [2].
>    Thus, the disentanglement results in Table 2—not accuracy—are central to our paper’s positioning. We acknowledge this was not clearly conveyed and will revise the discussion accordingly.
>
> 3. **On the nature and interpretability of emergent representations**:
>    Thank you for raising this. Topsim measures structural alignment between representation and semantic space, but not characteristics of the emergent language itself.
>    Following reviewer rCnw’s suggestion, we added metrics like Synonymy and Homonymy, and discussed them in the supplement (see also Weakness 4.i in their rebuttal). Our focus is on basic compositional properties; deeper interpretability metrics are outside the paper’s scope but worth future exploration.
>    Appendix I already shows how initial submessages carry relevant information—captured by our useful length metric. We will also provide more qualitative examples to support this.
>
> ---
>
> ### Questions
>
> 1. Thank you for pointing this out. We agree the original equation was overly condensed and have revised the notation for clarity.
>
> 2. Yes, $C$ is fixed and large enough that possible messages far outnumber dataset states. This ensures meaningful compression in useful length (see line 221) without harming reconstruction accuracy. We will clarify this in the text.
>
> 3. In line 134, the sentence “$\lambda > 1$ achieves the best performance” is a typo—should be “$\lambda \approx 1.5$.” Eq. (1) allows $\lambda > 0$, not just $\geq 1$ as the reviewer inferred (see Figure 2).
>    Also, $C$ is fixed in Eq. (1). As $\lambda$ increases, more weight is placed on later tokens. In the limit $\lambda 	o \infty$, PD becomes equivalent to full message reconstruction. Even when $\lambda = 1$, earlier tokens still contribute to more terms than later ones and are thus more heavily pressured. We will clarify this.
>
> Once again, we thank the reviewer for their time and valuable feedback.
>
> ---
>
> ### Shapes3D (10 runs)
>
> | Category           | Method                                        | TopSim ↑       | l₂×10⁻¹ ↓      | Last symbol MSE ↓ |
> |-------------------|-----------------------------------------------|----------------|----------------|-------------------|
> | **Interaction**    | Full message reconstruction, no IL (Baseline) | 0.244 ± 0.0021 | 10.0 ± 0.0     | 0.212 ± 0.003     |
> |                   | Progressive decoding (PD, ours)               | 0.270 ± 0.001  | 7.5 ± 0.215    | 0.238 ± 0.007     |
> | **Imitation**      | Message imitation (baseline)                  | 0.257 ± 0.005  | 9.9 ± 0.3      | 0.200 ± 0.007     |
> |                   | Full state imitation                          | 0.256 ± 0.003  | 9.9 ± 0.3      | 0.179 ± 0.003     |
> |                   | Final state imitation (FiSI, ours)            | 0.283 ± 0.003  | 10.0 ± 0.0     | **0.176 ± 0.02**  |
> | **Regularization** | PD+FiSI+KoLeo λ=1.5                           | 0.256 ± 0.002  | 8.555 ± 0.167  | 0.179 ± 0.003     |
> |                   | PD+FiSI+PDM (ours) λ=1.5                      | **0.292 ± 0.002** | **7.0 ± 0.155** | 0.194 ± 0.005     |
>
> ---
>
> ### MPI3D (10 runs)
>
> | Category           | Method                                        | TopSim ↑       | l₁.95×10⁻² ↓   | Last symbol MSE ↓ |
> |-------------------|-----------------------------------------------|----------------|----------------|-------------------|
> | **Interaction**    | Full message reconstruction, no IL (Baseline) | 0.133 ± 0.001  | 9.3 ± 0.002    | **0.015 ± 0.0**   |
> |                   | Progressive decoding (PD, ours)               | 0.137 ± 0.001  | **6.6 ± 0.341**| **0.015 ± 0.0**   |
> | **Imitation**      | Message imitation (baseline)                  | 0.135 ± 0.001  | 9.733 ± 0.029  | 0.016 ± 0.0       |
> |                   | Full state imitation                          | 0.137 ± 0.001  | 9.923 ± 0.020  | 0.018 ± 0.0       |
> |                   | Final state imitation (FiSI, ours)            | **0.156 ± 0.001** | 9.7 ± 0.046 | 0.02 ± 0.0        |
> | **Regularization** | PD+FiSI+KoLeo λ=1.5                           | 0.147 ± 0.002  | 8.9 ± 0.221    | 0.02 ± 0.0        |
> |                   | PD+FiSI+PDM (ours) λ=1.5                      | 0.153 ± 0.001  | 9.0 ± 0.167    | 0.02 ± 0.0        |
>
> ---
>
> [1] Zhenlin Xu, Marc Niethammer, and Colin Raffel. *Compositional generalization in unsupervised compositional representation learning: A study on disentanglement and emergent language*, 2022. [arXiv:2210.00482](https://arxiv.org/abs/2210.00482)
> [2] Milton Llera Montero et al. *The Role of Disentanglement in Generalisation*, ICLR 2021. [OpenReview](https://openreview.net/forum?id=qbH974jKUVy)

---

> > ### Comment · Area_Chair_2oGp · 2025-08-04
> > **AC Request**
> >
> > Dear Reviewer,
> >
> > Since the authors have provided additional experimental results, I would appreciate it if you could take their response into consideration and indicate whether it adequately addresses your concerns.
> >
> > Best,
> >
> > AC

---

> > ### Comment · Reviewer_Likm · 2025-08-05
> >
> > I thank the authors for their responsive rebuttal. Just to clarify, statistical significance in Table 1 is not the main issue in my view but rather the small effect size, especially without a deeper assessment of the nature of representations.
> >
> > I'm confused by the following remark:
> >
> > > Appendix I already shows how initial submessages carry relevant information—captured by our useful length metric.
> >
> > I have not seen any appendix in the submission. Am I missing something?

---

> > > ### Author Response · Authors · 2025-08-05
> > >
> > > We thank the reviewer for their response.
> > >
> > > While we agree the intermediate results for table 1 suggest a small effect size, the justification for our method relies on the joint use of our improvements: FiSI gives greater freedom for language drift between iterations, PDM (and entropy regularization in general) pushes the newly learned language to cover the space of possible messages more completely and further encourages exploration and drift, and PD serves to anchor the communication process, in effect reducing the amount of freedom the language has in the interaction phase. The combination of these additions in comparison to the initial baseline shows a meaningful impact over all metrics in table 1. We will make the importance of the combination of all additions more clear in the final work.
> > >
> > > For a deeper assessment of the results, please note that we have added the metrics of synonymy, homonymy, word order freedom and disentanglement as suggested by reviewer rCnw, as well as the original metrics for disentanglement and usefulness for downstream tasks (which we have updated with additional runs in the rebuttal to reviewer rCnw). Furthermore, as mentioned before, we plan to add more images for qualitative assessment, similar to Appendix I.
> > > As for the results already present in the appendix, we ask the reviewer to please refer to the supplementary material zip file associated with this submission.

---

> > > > ### Comment · Reviewer_Likm · 2025-08-05
> > > >
> > > > Ah, thanks for referring me to the supplemental zip file!
> > > >
> > > > I still have some of the reservations I mentioned in regards to the overall significance of the results, but given the authors responsive rebuttal I feel a bit more positively about this work and am inclined to increase a my rating.

---

### Decision · Program_Chairs · 2025-09-17

**Decision:**

Accept (poster)

**Comment:**

*Main Contribution*
The paper proposes a novel approach to compositional representation learning through a compressive–expressive communication framework. On two synthetic datasets, the method demonstrates improved disentanglement (i.e., compositional representation), albeit at the cost of reduced reconstruction performance.

*Strengths*
The paper addresses an interesting problem and provides sufficient methodological contributions. The results are clearly presented, and the authors thoughtfully discuss both the limitations of their work and its connections to related research.

*Weaknesses*
Reviewers raised concerns about the significance of the empirical results, noting that they appear marginal. Another issue is the trade-off inherent in the approach: the method improves compositionality but sacrifices reconstruction performance. More comprehensive experiments are needed to fully evaluate this trade-off. In addition, the method represents only a modest modification of existing techniques, and the connection to chain-of-thought prompting remains unclear.

*Rebuttal*
In their rebuttal, the authors provided additional experimental results, which convinced some reviewers to increase their scores and partially addressed concerns regarding the practical significance of the method.

*Justification of decision*
Following the rebuttal, the reviewers reached a consensus of “borderline acceptance.”